# Two-step detection of Lewy body pathology via smell-function testing and CSF α-synuclein seed amplification

Sophie E. Mastenbroek [1,2,3] ✉, Lyduine E. Collij [1,2,3], Jacob W. Vogel [4], Serena Caldera[5], Geidy E. Serrano[6], Charles H. Adler[7], Claudia Marina Vargiu [5], Sebastian Palmqvist [1,8], Frederik Barkhof [2,3,9], Piero Parchi [5,10], Thomas G. Beach[6], Rik Ossenkoppele [1,11,12] & Oskar Hansson [1] ✉

Cerebrospinal fluid (CSF) α-synuclein (α-syn) seed amplification assays (SAAs) can detect Lewy body pathology (LBP) with high accuracy but are invasive and costly. To address these challenges, this study evaluated a two-step workflow combining prescreening via smell-function testing with confirmatory CSF α-syn SAA testing only in individuals with reduced smell, for predicting *post-mortem* LBP status. Among 358 autopsied participants, the two-step workflow predicted brain LBP with high accuracy overall (94%), and within clinical subgroups (clinical parkinsonism=95%; clinical Alzheimer's disease [AD]=94%; clinically unimpaired [CU]=93%). It reduced the need for confirmatory CSF testing by 43% overall (23% clinical parkinsonism; 35% clinical AD; 80% CU). In an independent in vivo cohort (*N*=1209), the workflow predicted CSF α-syn SAA status with 79% accuracy and reduced CSF testing by 26%. This approach may reduce invasive CSF testing, alleviating patient burden and lowering healthcare costs.

Lewy body pathology (LBP), characterized by the intraneuronal aggregation of misfolded α-synuclein (α-syn), is a pathologic hallmark of Lewy body (LB) diseases (including Parkinson's disease [PD] and dementia with Lewy bodies [DLB], collectively referred to as neuronal α-syn disease[1]). Recently, the development of cerebrospinal fluid (CSF) seed amplification assays (SAAs) for the detection of α-syn seeds has provided an accurate in vivo biomarker of LBP[2], even during pro-dromal and preclinical stages[3,4]. In addition, the CSF α-syn SAA has been shown to be a valuable biomarker of comorbid LBP in Alzheimer's

disease (AD), where its presence has been associated with accelerated clinical progression[4,5]. However, CSF sampling is invasive and costly, and the SAA is very time-consuming, limiting widespread use.

To address these challenges, a two-step diagnostic workflow could be beneficial, incorporating a prescreening step to identify individuals at high risk of LBP who would then undergo lumbar puncture for CSF collection. Similar approaches have been proposed in AD, where plasma biomarkers are used to prescreen for amyloid-β or tau status, reducing confirmatory CSF or positron emission

[1]Clinical Memory Research Unit, Department of Clinical Sciences Malmö, Faculty of Medicine, Lund University, Lund, Sweden. [2]Department of Radiology and Nuclear Medicine, Vrije Universiteit Amsterdam, Amsterdam University Medical Center location VUmc, Amsterdam, The Netherlands. [3]Amsterdam Neuroscience, Brain imaging, Amsterdam, The Netherlands. [4]Department of Clinical Sciences Malmö, Faculty of Medicine, SciLifeLab, Lund University, Lund, Sweden. [5]IRCCS Istituto delle Scienze Neurologiche di Bologna, Bologna, Italy. [6]Banner Sun Health Research Institute, Sun City, AZ, USA. [7]Department of Neurology, Parkinson's Disease and Movement Disorders Center, Mayo Clinic, Scottsdale, AZ, USA. [8]Memory Clinic, Skåne University Hospital, Malmö, Sweden. [9]Institutes of Neurology & Healthcare Engineering, University College London, London, UK. [10]Department of Biomedical and Neuromotor Sciences, University of Bologna, Bologna, Italy. [11]Alzheimer Center Amsterdam, Neurology, Vrije Universiteit Amsterdam, Amsterdam UMC location VUmc, Amsterdam, The Netherlands. [12]Amsterdam Neuroscience, Neurodegeneration, Amsterdam, The Netherlands. ✉e-mail: sophie.mastenbroek@med.lu.se; oskar.hansson@med.lu.se

tomography tests by up to 80%[6–8]. In the context of LBP, the University of Pennsylvania Smell Identification Test (UPSIT), a measure of olfactory dysfunction, is a promising prescreening tool due to its low cost, accessibility, correlation to LBP burden, and demonstrated accuracy in differentiating PD and DLB from other neurodegenerative diseases[9–12].

In this study, we aimed to evaluate the performance of a two-step workflow for accurately predicting *postmortem* LBP status while limiting the number of lumbar punctures needed. This was performed in a heterogenous longitudinal cohort ($N = 358$) with *antemortem* UPSIT scores, *postmortem* CSF α-syn SAA results, and *postmortem* neuropathological assessments of regional LBP load. In step 1, a risk stratification model predicting *postmortem* LBP status was developed using 5-fold cross-validated logistic regression models with UPSIT, age, and sex as predictors across 1000 iterations. In step 2, results from confirmatory CSF SAA testing were only used in participants with elevated risk according to step 1 (Fig. 1a). The primary analysis included all participants, with sub-analyses in (i) patients presenting with clinical

parkinsonism; (ii) those with AD-related clinical symptoms; and (iii) clinically unimpaired (CU) individuals. Three neuropathological reference standards were used: (i) the presence (or not) of LBP in any brain region, (ii) the presence (or not) of LBP in any cortical brain region, and (iii) the presence (or not) of LBP in brainstem and limbic or neocortical brain regions. To confirm that the use of *postmortem* CSF did not introduce bias, we further validated the two-step workflow in an independent cohort with in vivo CSF α-syn SAA data ($N = 1209$).

## Results

358 neuropathological samples with *antemortem* UPSIT scores (ranging from 0 to 40 with higher scores indicating better olfactory function) and *postmortem* ventricular brain CSF were selected from the Arizona Study of Aging and Neurodegenerative Disorders (AZSAND)/ Brain and Body Donation Program (BBDP)[13]. The mean age at death was $86.2 \pm 7.8$ years, 42.7% was female, mean *postmortem* interval (PMI) was $3.9 \pm 3.9$ h, and the mean interval between UPSIT test and death

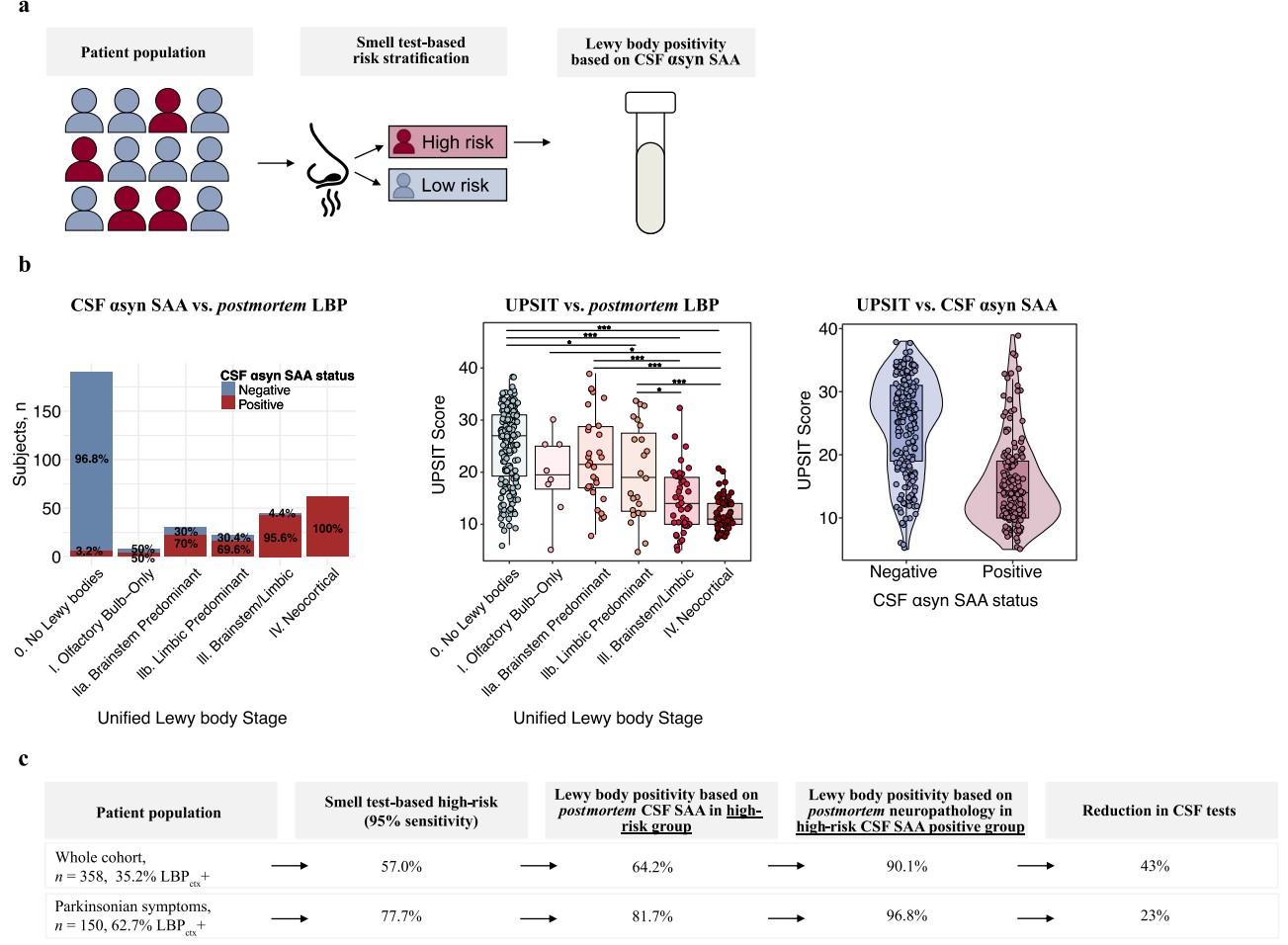

**Fig. 1 | Two-step workflow design and results summary. a** Design of a conditional two-step workflow to detect Lewy body pathology (LBP). Step 1 consists of UPSIT-based risk stratification into high- and low-risk groups having cortical LBP (LBP$_{ctx}$). Step 2 includes confirmatory CSF α-syn seeding amplification assay (SAA) testing in high-risk patients identified in step 1. **b** The association between *postmortem* LBP, *postmortem* CSF α-syn SAA, and UPSIT scores ($n = 358$). **c** Summary of the proportion of individuals selected as high−risk, LB-positive based on *postmortem* CSF, LB-positive based on *postmortem* neuropathology in the cortex, and the reduction in CSF tests, for each of the four scenarios. Boxplots show the median, lower, and upper quartiles with whiskers representing minimum and maximum values. Source data are provided as a Source Data file. Comparisons were performed with two-sided linear regression models adjusted for multiple comparisons (tukey method). * $p < 0.05$; ** $p < 0.01$; *** $p < 0.001$. αsyn alpha-synuclein, CSF cerebrospinal fluid, ctx cortex, LBP Lewy body pathology, SAA seed amplification assay, UPSIT University of Pennsylvania Smell Identification Test.

was $3.2 \pm 2.3$ years (Table S1). The cohort comprised a wide variety of neurodegenerative diseases, with AD and PD being the most common clinicopathological diagnoses (Table S2; *antemortem* clinical diagnosis in Table S3).

42.5% was CSF α-syn SAA-positive (+), 35.2% was LB+ based on cortical LBP (LBP$_{ctx}$; neuropathology; LBs in at least one cortical brain region), 47.5% based on LBP in any brain region (LBP$_{any}$; neuropathology; LBs in at least one of all studied brain regions), and 29.9% based on brainstem/limbic or neocortical LBP (LBP$_{B/L-N}$; neuropathology; corresponding to Unified Staging System for Lewy body Disorders [USSLB] stages III. Brainstem/Limbic or IV. Neocortical) (Table S1). Among three individuals with multiple system atrophy (MSA) (Table S2), none showed evidence of *postmortem* LBP in any of the 10 tested brain regions, while one individual tested positive on the CSF α-syn SAA.

The proportion of individuals classified as CSF α-syn SAA+ increased with more widespread *postmortem* LBP as identified by the USSLB (stage I Olfactory Bulb-Only = 50.0%, stage IIa Brainstem Predominant = 71.0%, stage IIb Limbic Predominant = 69.6%, stage III Brainstem/Limbic = 95.6%, and IV Neocortical = 100.0%) (Fig. 1b). Among those without *postmortem* LBP (USSLB stage 0), 96.8% were CSF α-syn SAA LB-negative (-). UPSIT scores decreased with more advanced *postmortem* LBP. Individuals classified as LBP+ based on neuropathology (LBP$_{ctx}$+, LBP$_{any}$+, or LBP$_{B/L-N}$+) and individuals with a positive CSF α-syn SAA result had lower UPSIT scores, reflecting worse olfactory function (Fig. 1b, Fig. S1).

## UPSIT-based risk stratification for presence of LBP$_{ctx}$ (step 1)

In step 1, a risk stratification model predicting *postmortem* LBP$_{ctx}$ status was developed in the complete cohort using 5-fold cross-validated logistic regression models with UPSIT, age, and sex as predictors (Table S4). Individual risk probabilities were derived from this model and four different thresholding strategies, corresponding to 80%, 85%, 90%, and 95% sensitivity, were explored to classify participants into groups with low and high risk of LBP$_{ctx}$-positivity. In the whole cohort, the size of the UPSIT-based "high-risk group" increased with more lenient probability thresholds (i.e., higher sensitivity levels) (Table 1, Fig. S2), resulting in fewer false negatives (i.e., fewer missed LBP$_{ctx}$+ individuals) (Fig. 1C). Similar findings were observed in three clinical subgroups of (i) individuals with clinical parkinsonian symptoms ($n = 150$); (ii) individuals with clinical symptoms of AD ($n = 97$); and (iii)

CU individuals ($n = 44$). We considered a low false negative rate most important given the prescreening purpose of the UPSIT-based risk stratification. Hence, we selected the most inclusive probability threshold (95% sensitivity) for the primary analyses and show results of more stringent thresholds in Table S5.

## CSF α-syn SAA testing in high-risk individuals for detection of LBP$_{ctx}$ (step 2)

In step 2, CSF α-syn SAA testing was restricted to the high-risk group. To assess the performance of the two-step workflow, we computed the overall workflow accuracy, positive predictive value (PPV) and negative predictive value (NPV), as well as the reduction in the number of CSF tests needed. Restricting CSF α-syn SAA testing to the high-risk group substantially reduced the number of CSF tests required (Fig. 2d), while maintaining high accuracy, PPV, and NPV across the whole cohort and within the three clinical subgroups (Fig. 2a–c, Table S5). The most significant reduction in CSF tests was observed in CU individuals (80%). Accuracy was comparable across clinical scenarios (93–95%), with the highest PPV in clinical parkinsonism (96%) and the highest NPV in clinical AD (98%). The two-step diagnostic workflow-maintained performance compared to CSF α-syn SAA testing in every subject, while substantially outperforming UPSIT-based risk stratification alone. This improvement was most pronounced in the clinical AD subgroup (Fig. 2).

## Sensitivity analyses

Using *postmortem* LBP$_{any}$ as the reference standard (instead of LBP$_{ctx}$) yielded slightly lower accuracies and NPVs, but higher PPVs (Tables S6, S7). A lower reduction in CSF tests was observed (4.6–38.6%) (Fig. S3A). Using LBP$_{B/L-N}$ as the reference yielded highly comparable accuracies, higher NPVs, and lower PPVs than LBP$_{ctx}$ (Tables S8, S9) and slightly more CSF tests were saved (26.7–84.1%) (Fig. S3B), Limiting analyses to individuals who underwent UPSIT testing within 5 years of death did not affect model performance (Table S10).

To examine whether the use of *postmortem* CSF influenced the results, we also validated the two-step workflow in an independent cohort (i.e., the Parkinson's Progression Markers Initiative [PPMI] cohort) consisting of 1209 individuals with in vivo CSF α-syn SAA and UPSIT available. Mean age was $65.1 \pm 8.7$ years, 46.8% was female, and 68.8% was CSF α-syn SAA+ (Table S11). The majority had a PD diagnosis (55.7%) or was a healthy control (38.9%) (Table S11). We applied the

**Table 1 | Model-based risk stratification for cortical LBP (LBP$_{ctx}$) positivity**

| | Whole cohort | | | Clinical parkinsonism | | | Clinical AD | | | Clinically unimpaired | | |
|---|---|---|---|---|---|---|---|---|---|---|---|---|
| | All | LBP$_{ctx}$- | LBP$_{ctx}$+ | All | LBP$_{ctx}$- | LBP$_{ctx}$+ | All | LBP$_{ctx}$- | LBP$_{ctx}$+ | All | LBP$_{ctx}$- | LBP$_{ctx}$+ |
| *80% sensitivity* | | | | | | | | | | | | |
| Low risk | 213 | 190 (89.2) | 23 (10.8) | 57 | 47 (82.5) | 10 (17.5) | 54 | 49 (89.3) | 5 (10.7) | 42 | 38 (90.5) | 4 (9.5) |
| High risk | 143 | 41 (28.7) | 102 (71.3) | 93 | 9 (9.7) | 84 (90.3) | 41 | 22 (53.7) | 21 (46.3) | 2 | 1 (50.0) | 1 (50.0) |
| *85% sensitivity* | | | | | | | | | | | | |
| Low risk | 203 | 183 (90.1) | 20 (9.9) | 52 | 44 (84.6) | 8 (15.4) | 49 | 45 (91.8) | 4 (8.2) | 42 | 38 (90.5) | 4 (9.5) |
| High risk | 157 | 49 (31.2) | 106 (68.8) | 98 | 12 (12.2) | 86 (87.8) | 48 | 26 (54.2) | 22 (45.8) | 2 | 1 (50.0) | 1 (50.0) |
| *90% sensitivity* | | | | | | | | | | | | |
| Low risk | 187 | 173 (92.5) | 14 (7.5) | 47 | 43 (91.5) | 4 (7.0) | 45 | 43 (95.6) | 2 (4.4) | 39 | 35 (89.7) | 4 (10.3) |
| High risk | 171 | 59 (34.5) | 112 (65.5) | 103 | 13 (12.6) | 90 (87.4) | 52 | 28 (53.8) | 24 (46.2) | 5 | 4 (80.0) | 1 (20.0) |
| *95% sensitivity* | | | | | | | | | | | | |
| Low risk | 154 | 148 (96.1) | 6 (3.9) | 35 | 34 (97.1) | 1 (2.9) | 33 | 32 (97.0) | 1 (3.0) | 35 | 33 (94.3) | 2 (5.7) |
| High risk | 214 | 84 (39.3) | 120 (60.7) | 115 | 22 (19.1) | 93 (80.9) | 64 | 39 (60.9) | 25 (38.1) | 9 | 6 (66.7) | 3 (33.3) |

Data are presented as *n* or *n* (%). The first column indicates the evaluated strategies with different sensitivity-based thresholds for UPSIT-derived risk stratification. For each strategy, the total number of individuals in the low- and high-risk groups are shown, followed by numbers of Lewy body pathology negative (LBP-) and LBP+ participants according to *postmortem* cortical neuropathology measures (ctx). The percentage of LBP$_{ctx}$-negatives in the low-risk group and the percentage of LBP$_{ctx}$-positives in the high-risk group correspond to each evaluated threshold's NPV and PPV, respectively.
*AD* Alzheimer's disease, *ctx* cortex, *LBP* Lewy body pathology.

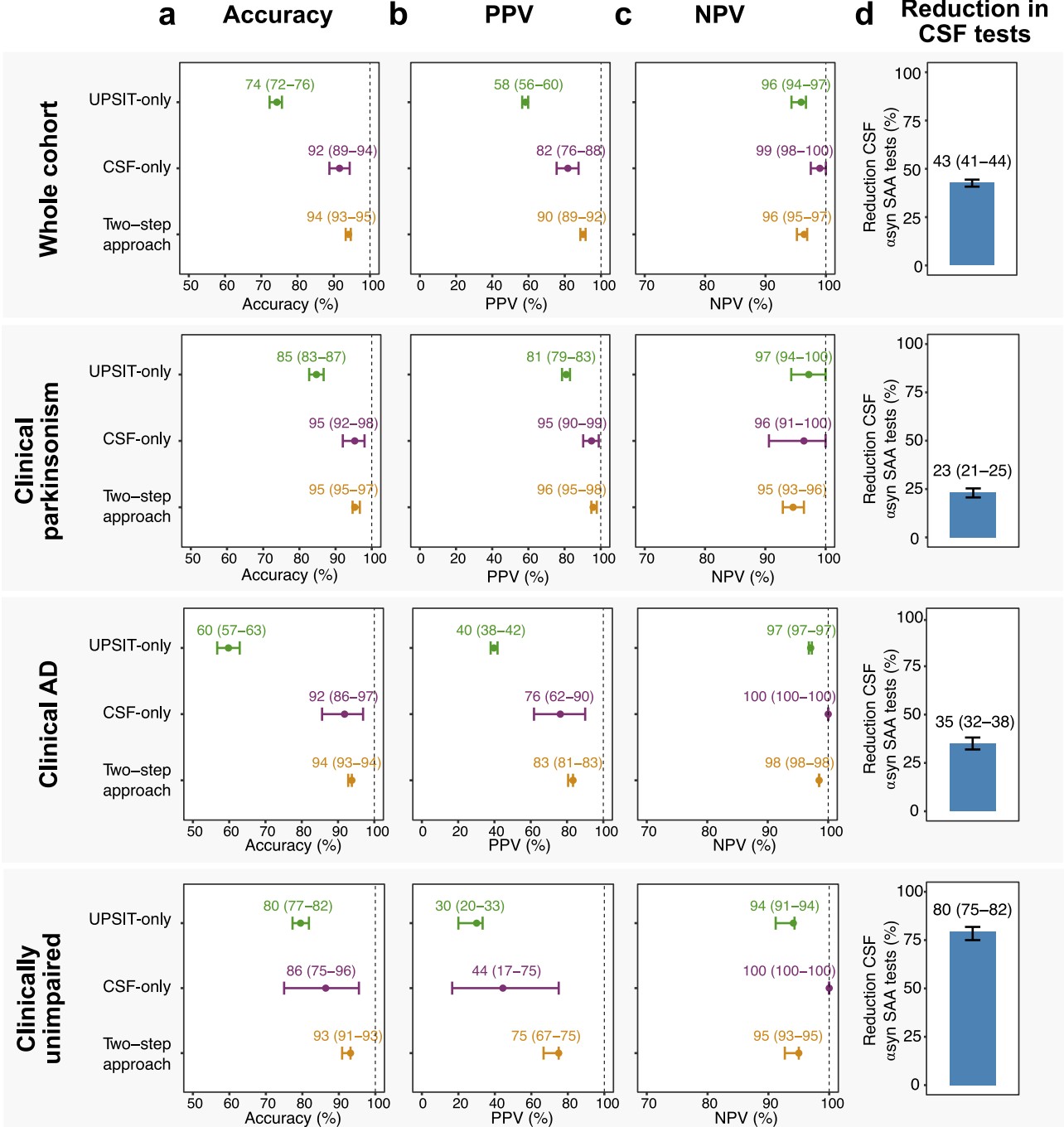

**Fig. 2 | Performance of a two-step workflow for detecting cortical LB pathology.** Performance and reduction in CSF α-syn SAA tests across the whole cohort and subgroups of individuals with clinical signs of parkinsonism, clinical Alzheimer's disease, and clinically unimpaired individuals. For illustrative purposes, probabilities and datapoints correspond to the median of 1000 iterations and thresholds corresponding to 95% sensitivity were used. **a–c** The median accuracy, positive predictive value (PVV) and negative predictive value (NPV) of the two-step, CSF-only, and UPSIT-only approaches. **d** The observed median percentage of reduction in CSF tests using the two-step workflow. Error bars correspond to 95% CIs based on 1000 iterations of model development and classification. αsyn alpha-synuclein, AD Alzheimer's disease, CSF cerebrospinal fluid, NPV negative predictive value, PPV positive predictive value, SAA seed amplification assay, UPSIT University of Pennsylvania Smell Identification Test.

UPSIT-based risk stratification model from the autopsy cohort to the in vivo cohort, where it yielded robust predictions of CSF α-syn SAA status (accuracy = 79%; PPV = 82%; NPV = 69%), while reducing the number of CSF tests required by 26% (Fig. 3).

## Discussion

We show that a two-step diagnostic workflow, combining UPSIT-based risk-stratification (step 1) and restricting confirmatory CSF testing to high-risk individuals identified in step 1 (step 2), can accurately predict *postmortem* LBP status in a heterogeneous cohort while reducing the number of necessary lumbar punctures (Fig. 1).

Our proposed two-step diagnostic workflow has potential utility across various clinical scenarios where CSF α-syn SAA testing might be relevant. Among individuals with clinical parkinsonism, the UPSIT by itself already yielded high performance (accuracy = 85%), which improved to 95% with the addition of step 2. However, the number of

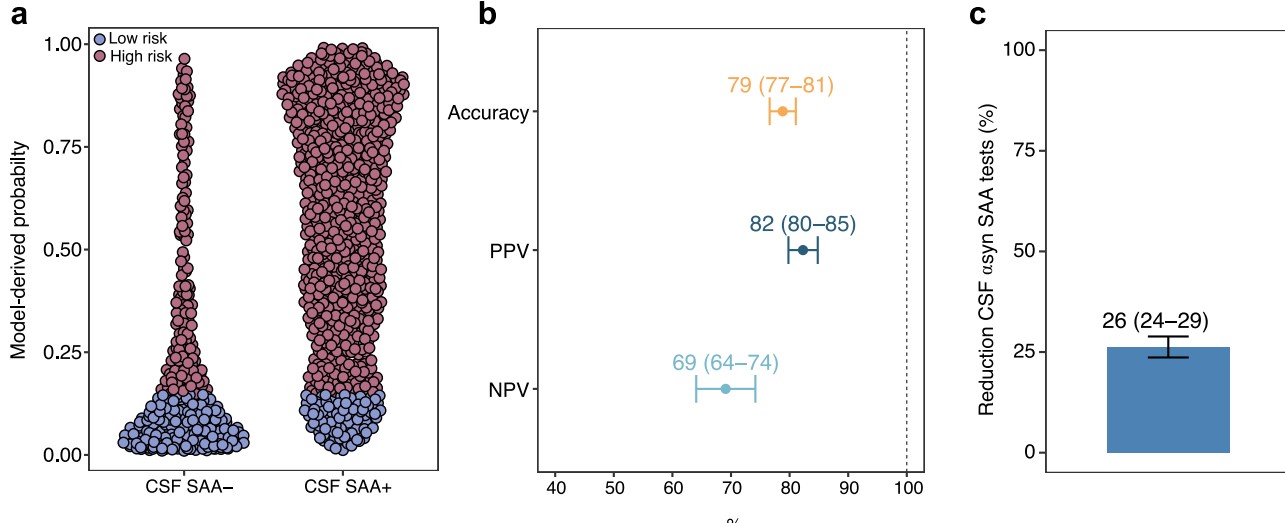

**Fig. 3 | Application of the two-step workflow to an in vivo clinical cohort.** The two-step workflow applied to an in vivo clinical cohort. **a** Distribution of model-derived probabilities for cortical Lewy body pathology (LBP) based on a logistic regression model including UPSIT scores, age, and sex as predictors, trained on the autopsy dataset. A probability threshold corresponding to 95% sensitivity, derived from the autopsy dataset, was used to classify individuals as low- (blue dots) or high- (red dots) risk for cortical LBP. **b** Performance metrics of the UPSIT-based risk classification groups for predicting CSF α-syn SAA status, including accuracy, positive predictive value (PPV), and negative predictive value (NPV). **c** Reduction in the number of CSF tests required using the two-step workflow. Error bars correspond to 95% CIs based on 1000 iterations of model development and classification. 1209 PPMI participants were included in the analyses. αsyn alpha-synuclein, CSF cerebrospinal fluid, NPV negative predictive value, PPV positive predictive value, SAA seed amplification assay.

CSF tests saved in this group was modest (23%). In contrast, in individuals presenting with symptoms suggestive of AD, where *postmortem* studies have shown that comorbid LBP occurs in 30-40%[14], the UPSIT-based model (step 1) performed less well. This is likely due to the influence of other factors affecting performance on a test for smell function, such as age-related factors, cognitive impairment, or the accumulation of AD pathology in the olfactory system[15–17]. In this scenario, the two-step workflow improved LBP classification performance substantially over UPSIT alone, while reducing the number of lumbar punctures compared to a CSF-only approach. This is particularly relevant for AD clinical trials, where identifying comorbid pathologies such as LBP is critical for understanding their impact on clinical and biological disease progression. Finally, we show the potential of the UPSIT as a prescreening tool in clinically normal populations, where the number of confirmatory CSF tests was reduced by 80% while improving overall accuracy. This approach could be especially important for future clinical interventional trials targeting *preclinical* LBD pathology.

Our findings support previous autopsy-based studies demonstrating the high accuracy of CSF α-syn SAAs for identifying individuals with LBP[3,18–28]. In our study, the assay demonstrated excellent overall specificity (96.8%) and sensitivity (87.0%) in predicting autopsy-confirmed LBP. Consistent with previous research, sensitivity was highest in individuals with more advanced pathology, reaching 95.6% in cases with brainstem and limbic involvement and 100% in neocortical LBP. Conversely, sensitivity was lower in individuals with focal (early-stage) disease (i.e., 50.0% olfactory bulb and ≈70.0% limbic- or brainstem-predominant). These values align with previous studies reporting sensitivities ranging from 14.3% to 63.6% for brainstem- and amygdala-predominant LBP, and from 90% to 100% for limbic and neocortical LBP[18,22–26,28]. Accordingly, our proposed two-step diagnostic workflow yielded higher accuracy in predicting cortical or brainstem/limbic or neocortical *postmortem* LBP status compared to predicting the presence of any LBP. The reduced sensitivity of the assay in early-stage cases has been hypothesized to reflect either a lower pathological burden or the presence of a distinct pathological

strain of α-syn that may be less well detected by the assay[25]. Future research should aim to increase sensitivity in focal LBP to improve early detection and clinical applicability.

It should be noted that olfactory impairment is not specific to Lewy body disease and has been observed across a range of other neurodegenerative diseases, including AD, multiple sclerosis, amyotrophic lateral sclerosis, and Huntington's disease[29–32]. In addition, olfactory function can be affected by several non-degenerative factors, as the nasal neuroepithelium is in direct communication with the external environment. For instance, smoking, head trauma, and chronic sinonasal diseases have all been associated with reduced olfactory function (see Table S12 for the frequencies of these factors in our cohort)[32]. Factors influencing smell function might confound the interpretation of the UPSIT results in the context of Lewy body disease, potentially leading to false positives but not false negatives. In our proposed two-step workflow, the inclusion of the CSF α-syn SAA test as a confirmatory measure might help mitigate this limitation, as shown by the increased accuracy of the two-step approach compared to using the UPSIT alone.

Strengths of this study include the composition of the dataset, which features a broad range of clinical diagnoses beyond Lewy body disease, allowing for robust evaluation of the two-step diagnostic workflow across different clinical groups, as well as replication in an in vivo cohort. This study also has several limitations. First, replication in independent cohorts with *postmortem* validation is needed, although there are very few cohorts in the world featuring *antemortem* smell testing, CSF sampling, and detailed *postmortem* neuropathological assessments. Second, the interval between *antemortem* UPSIT testing and autopsy was relatively long. However, analyses limited to participants with a maximum 5-year interval did not affect the performance of the two-step workflow. Finally, while the use of *postmortem* CSF samples offers the advantage of collection at the same point in time as the neuropathological assessment, several potential limitations should be considered. First, it has been speculated that *postmortem* CSF may be affected by overall instability or protein degradation. However, it should be noted that the current cohort has

drawn CSF samples from subjects with a short *postmortem* interval (PMI; median = 3.1 h). Multiple studies, including our own, have demonstrated that *postmortem* CSF yields results comparable to *antemortem* CSF regarding analyses such as Western blot, ELISA, proteomic, and metabolomic methodology[33–38]. Second, concerns have been raised about the equivalence of ventricular versus lumbar CSF. However, a comprehensive proteomics study comparing ventricular and lumbar CSF from the same individuals observed significant differences in protein levels for only two proteins[39]. Third, prior studies using autopsy-confirmed *antemortem* CSF α-syn SAA testing reported sensitivity and specificity values for detecting LBP that are similar to those observed in the current study[3,19–28]. Notably, one study directly compared the sensitivity and specificity of CSF α-syn SAA testing using both *antemortem* and *postmortem* samples from the same individuals, and found overall sensitivity to be slightly higher in *postmortem* CSF (80% vs. 71.2%), possibly because *postmortem* CSF was collected closer in time to autopsy than *antemortem* CSF, while specificity was slightly higher in *antemortem* CSF (98.1% vs. 88.5%)[22]. Importantly, we show that the use of *postmortem* CSF did not substantially affect the results, as the two-step workflow accurately predicted CSF SAA status in an independent in vivo cohort. While these findings suggest that *postmortem* CSF SAA results may generally be comparable to those obtained from *antemortem* samples, further research is needed to fully understand the differences. In addition, future research should explore the use of age- and sex-adjusted UPSIT percentiles to improve clinical applicability.

In summary, we present an accurate two-step approach for predicting LBP using a smell test followed by CSF α-syn SAA testing in smell test-positive individuals. This could minimize costs, reduces patient burden, and improve the known underdiagnosis of DLB and PD[5].

## Methods

### Participants

Neuropathological samples were selected from the Arizona Study of Aging and Neurodegenerative Disorders (AZSAND)/Brain and Body Donation Program (BBDP), which was approved by the SHRI Institutional Review Board[13]. All enrolled subjects signed an Institutional Review Board-approved informed consent. We included all autopsied BBDP participants with *antemortem* UPSIT scores and *postmortem* ventricular brain CSF of sufficient quality ($N$ = 358), excluding 9 samples with visible blood contamination or insufficient remaining CSF volume. The UPSIT assesses olfactory function by requiring subjects to identify 40 odorants in a multiple-choice format, using "scratch and sniff" labels. The UPSIT was administered by a trained technician and scored using standard procedures. Scores range from 0 to 40, with lower scores indicating worse olfactory function[12].

Detailed information about the PPMI study (ClinicalTrials.gov, number NCT01141023) can be found on the PPMI website (https://www.ppmi-info.org) and in previous publications[40,41]. We included all individuals with UPSIT scores and CSF α-syn SAA testing, excluding those with a genetic PD mutation and those who were recruited into the prodromal cohort, as their inclusion was based on hyposmia, to avoid bias introduced by UPSIT-driven recruitment ($N$ = 1209). CSF α-syn SAA testing was performed with the Amprion assay, following a detailed protocol[42,43]. Importantly, while different CSF SAAs were used in the PPMI and AZSAND/BBDP cohorts, a previous study directly comparing the two assays in the same individuals demonstrated remarkably similar results[44], consistent with findings from other studies comparing different assay versions[45].

### *Postmortem* assessments in the AZSAND/BBDP cohort

Brain harvesting, tissue processing, and staining protocols have been described in detail previously[13]. Immunohistochemical α-syn stainings were performed with a polyclonal antibody raised against an α-syn peptide fragment phosphorylated at serine 129 (pS129)[46,47]. Ten standard brain regions spanning the brainstem, limbic system, and neocortex (olfactory bulb and tract [OBT], anterior medulla, anterior and mid-pons, substantia nigra, mid-amygdala, transentorhinal area, anterior cingulate gyrus, middle temporal gyrus, middle frontal gyrus, and inferior parietal lobule) were sampled and graded for Lewy-type α-syn density according to a semi-quantitative rating scale ranging from 0 to 4 (0 = none, 1 = mild, 2 = moderate, 3 = severe, and 4 = very severe pathology)[48,49]. Participants were classified according to the Unified Staging System for Lewy Body Disorders (USSLB)[3,8] as 0. No Lewy bodies; I. Olfactory Bulb-Only; IIa. Brainstem Predominant; IIb. Limbic Predominant; III. Brainstem and Limbic; and IV. Neocortical. In addition, participants were classified as LB-/LB+ in the cortex (LBP$_{ctx}$, at least mild pathology [density score > 0] in transentorhinal area, anterior cingulate gyrus, middle temporal gyrus, middle frontal gyrus, or inferior parietal lobule), in any brain region (LBP$_{any}$, at least mild pathology [density score > 0] in any brain region), and in brainstem and limbic or neocortical brain regions (LBP$_{B/L-N}$, USSLB stage III or IV). Individuals were assigned a neuropathological diagnosis after death according to specific diagnostic criteria[13,49–56].

*Postmortem* CSF was collected from the lateral ventricles through the corpus callosum prior to brain removal, using 30 mL disposable polypropylene syringes fitted with 8 cm long, 18 gauge needles[13]. The CSF was ejected into 15 mL disposable polypropylene tubes for centrifugation and supernatants were aliquoted into 0.5 mL polypropylene microcentrifuge tubes and stored frozen at −80 °C. Next, we performed the CSF α-syn seeding amplification assay (SAA), including the purification of recombinant wild-type human αSyn, as previously described with minor analysis modifications[5,21]. For the SAA assay we used Black 96-well plates with a clear bottom (Nalgene Nunc International) pre-loaded with six 0.8 mm silica beads (OPS Diagnostics) per well. After thawing CSF samples and vortexing them for 10 s, 15 μl of CSF were added to 85 μl of reaction mix, containing 40 mM PB pH 8.0, 170 mM NaCl, 10 μM thioflavin-T (ThT), 0.0015% sodium dodecyl sulfate (SDS) and 0.1 mg/ml of wild-type recombinant α-syn filtered using a 100 kDa MWCO filter (Amicon centrifugal filters, Merck Millipore). The plate was sealed with a plate sealer film and incubated into a FLUOstar Omega (BMG Labtech) plate reader at 42 °C with intermittent double orbital shaking at 400 rpm for 1 min, followed by 1-min rest. ThT fluorescence measurements were taken every 45 min, using 450 nm excitation and 480 nm emission filters. We ran at least one positive and negative control on each plate. As positive controls, we used brain homogenates (10% in PBS) from areas with LB pathology, diluted 10-5 in a pool of αSyn negative CSF samples collected from patients diagnosed with normal pressure hydrocephalus (NPH). The same NPH samples were used as negative controls. Each post-mortem CSF sample was run undiluted and diluted 1:10. Samples were deemed positive when at least three out of four replicates reached a threshold arbitrarily set at 30% of the median of the maximum fluorescence intensity (Imax) reached by the positive control replicates. To minimize the risk of false-positive results, the analysis was repeated three times for those samples whose seeding activity was shown in only one or two of the four replicates in the first run. Ultimately, the result was considered "positive" when at least 4 of the 12 total replicates exceeded the threshold.

### Outcomes

The main outcome was LBP-positivity defined as having Lewy bodies in at least one of the 5 studied cortical brain regions (LBP$_{ctx}$+). The secondary outcomes were LBP-positivity defined as (i) having Lewy bodies in at least one of all 10 studied brain regions (LBP$_{any}$+), and (ii) having a USSLB stage III. Brainstem/Limbic or IV. Neocortical (LBP$_{B/L-N}$).

## Definition of clinical subgroups

To divide individuals across three different clinical scenarios, we used clinical summary data. Individuals were grouped into "clinical parkinsonism" when PD or DLB were stated in the clinical summary, or when another movement disorder was mentioned (e.g., multiple system atrophy, progressive supranuclear palsy, corticobasal degeneration), or when a symptom associated with parkinsonism was mentioned (e.g., parkinsonism, rapid eye movement behavior disorder, tremor, tremor disorder, restless leg syndrome). "Alzheimer's disease clinical symptoms" was assigned when the clinical summary mentioned Alzheimer's disease or amnestic MCI or dementia. When both parkinsonism and Alzheimer's disease were suspected, individuals were assigned to both groups. When no symptoms where mentioned, individuals were assigned to the "clinically unimpaired" group.

## Statistical analyses

*Postmortem* LBP was used as the reference standard, with LBP$_{ctx}$-status used in the main analyses and LBP$_{any}$- and LBP$_{B/L-N}$-status for sensitivity analyses. Logistic regression models predicting *postmortem* LBP status were developed using UPSIT score, age at the time of UPSIT testing, and sex as predictors. We chose to add age and sex as independent predictors, rather than using normative scores, as such scores are often derived from clinically defined control groups, which may include individuals with undetected LBP or other brain pathologies. Five-fold cross-validation was performed to derive individual prediction probabilities of LBP-positivity. Probability thresholds corresponding to 80%, 85%, 90%, and 95% sensitivity were applied to stratify participants into low- and high-risk groups. The performance of the two-step workflow was evaluated by testing the scenario in which confirmatory CSF α-syn SAA testing was performed only in high-risk participants. We computed the overall workflow accuracy, PPV and NPV, as well as the reduction in the number of CSF tests needed. These performance metrics were compared to single-step approaches using UPSIT-based risk stratification or CSF α-syn SAA testing alone. UPSIT-model development and risk classification were repeated across 1000 iterations, splitting the data into different folds each time. The median performance of the models (accuracy, PPVs and NPVs) is reported with 95% confidence intervals calculated from 1000 iterations.

To test two-step workflow performance across different clinical scenarios, performance was assessed in the entire cohort and in three clinical subgroups: individuals with clinical parkinsonian symptoms ($n = 150$), individuals with clinical symptoms of AD ($n = 97$), and CU individuals ($n = 44$).

Finally, we applied the UPSIT-, age-, and sex-based logistic regression model that was trained on the full AZSAND/BBDP cohort, to the PPMI cohort, to categorize individuals into low- and high-risk groups for cortical LBP. We then estimated the performance (accuracy, PPV, and NPV) of the UPSIT-based risk classification for predicting CSF α-syn SAA status, and the reduction of required CSF tests by using the UPSIT as a prescreening measure.

## Reporting summary

Further information on research design is available in the Nature Portfolio Reporting Summary linked to this article.

## Data availability

Anonymized data from the Arizona Study of Aging and Neurodegenerative Disorders/Brain and Body Donation Program will be shared by request as long as data transfer is in agreement with USA legislation (Privacy Rule of the Health Insurance Portability and Accountability Act). Data used in the preparation of this article was obtained on [2025-05-05] from the Parkinson's Progression Markers Initiative (PPMI) database (www.ppmi-info.org/access-data-specimens/download-data), RRID:SCR_006431. Raw data are available upon request. For up-to-date information on the study, visit www.ppmi-info.org. This analysis used DaTscan and αSyn-SAA results for prodromal participants, obtained from PPMI upon request after approval by the PPMI Data Access Committee. Source data are provided with this paper.

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

## Acknowledgements

Data used in the preparation of this article was obtained on [2025-05-05] from the Parkinson's Progression Markers Initiative (PPMI) database (www.ppmi-info.org/access-data-specimens/download-data), RRID:SCR_006431. For up-to-date information on the study, visit www.ppmi-info.org. Work at Lund University was supported by the European Research Council (ADG-101096455), Alzheimer's Association (ZEN24-1069572, SG-23-1061717), GHR Foundation, Swedish Research Council (2022-00775), ERA PerMed (ERAPERMED2021-184), Knut and Alice Wallenberg foundation (2022-0231), Strategic Research Area MultiPark (Multidisciplinary Research in Parkinson's disease) at Lund University, Swedish Alzheimer Foundation (AF-980907), Swedish Brain Foundation (FO2021-0293), Parkinson foundation of Sweden (1412/22), Cure Alzheimer's fund, Rönström Family Foundation, Konung Gustaf V:s och Drottning Victorias Frimurarestiftelse, Skåne University Hospital Foundation (2020-O000028), Regionalt Forskningsstöd (2022-1259) and Swedish federal government under the ALF agreement (2022-Projekt0080). The funding sources had no role in the design and conduct of the study; in the collection, analysis, interpretation of the data; or in the preparation, review, or approval of the manuscript. Work at IRCCS Istituto delle Scienze Neurologiche was supported by the Italian Ministero

della Salute (grant RF-2021-12374386). The Brain and Body Donation Program is supported by the National Institute of Neurological Disorders and Stroke (U24 NS072026 National Brain and Tissue Resource for Parkinson's Disease and Related Disorders), the National Institute on Aging (P30 AG19610 and P30 AG072980, Arizona Alzheimer's Disease Core Center), the Arizona Department of Health Services (contract 211002, Arizona Alzheimer's Research Center), the Arizona Biomedical Research Commission (contracts 4001, 0011, 05-901 and 1001 to the Arizona Parkinson's Disease Consortium) and the Michael J. Fox Foundation for Parkinson's Research. PPMI – a public-private partnership—is funded by the Michael J. Fox Foundation for Parkinson's Research and funding partners, including 4D Pharma, Abbvie, AcureX, Allergan, Amathus Therapeutics, Aligning Science Across Parkinson's, AskBio, Avid Radio-pharmaceuticals, BIAL, BioArctic, Biogen, Biohaven, BioLegend, Blue-Rock Therapeutics, Bristol-Myers Squibb, Calico Labs, Capsida Biotherapeutics, Celgene, Cerevel Therapeutics, Coave Therapeutics, DaCapo Brainscience, Denali, Edmond J. Safra Foundation, Eli Lilly, Gain Therapeutics, GE HealthCare, Genentech, GSK, Golub Capital, Handl Therapeutics, Insitro, Jazz Pharmaceuticals, Johnson & Johnson Innovative Medicine, Lundbeck, Merck, Meso Scale Discovery, Mission Therapeutics, Neurocrine Biosciences, Neuron23, Neuropore, Pfizer, Piramal, Prevail Therapeutics, Roche, Sanofi, Servier, Sun Pharma Advanced Research Company, Takeda, Teva, UCB, Vanqua Bio, Verily, Voyager Therapeutics, the Weston Family Foundation and Yumanity Therapeutics.

## Author contributions

S.E.M., O.H., J.W.V., R.O., L.E.C., and F.B. designed the study. S.E.M. performed the analyses and data interpretations under the supervision of O.H., J.W.V., R.O., L.E.C., F.B., and S.P. The manuscript was drafted by S.E.M. All authors contributed to preparation and critical review of the manuscript. T.G.B., G.E.S., C.A., P.P., S.C., and C.M.V. collected the data.

## Funding

## Competing interests

O.H. is an employee of Eli Lilly and Lund University, and he has previously acquired research support (for Lund University) from AVID Radiopharmaceuticals, Biogen, C2N Diagnostics, Eli Lilly, Eisai, Fujirebio, GE Healthcare, and Roche. In the past 2 years, he has received consultancy/speaker fees from Alzpath, BioArctic, Biogen, Bristol Meyer Squibb, Eisai, Eli Lilly, Fujirebio, Merck, Novartis, Novo Nordisk, Roche, Sanofi and Siemens. L.E.C. has received research support from GE Healthcare (paid to institution). SP has acquired research support (for the institution) from Avid and ki elements through ADDF. In the past 2 years, he has received consultancy/speaker fees from Bioartic, Biogen, Esai, Eli Lilly, Novo Nordisk, and Roche. F.B. acts as a consultant for Biogen-Idec, IXICO, Merck-Serono, Novartis, Combinostics, and Roche. He has received grants, or grants are pending, from the Amyloid Imaging to Prevent Alzheimer's Disease (AMYPAD) initiative, the Biomedical Research Centre at University College London Hospitals, the Dutch MS Society, ECTRIMS–MAGNIMS, EU-H2020, the Dutch Research Council (NWO), the UK MS Society, and the National Institute for Health Research, University College London. He has received payments for the development of educational presentations from Ixico and his institution from Biogen-Idec and Merck. He is on the editorial board of Radiology, European Neuroradiology, Multiple Sclerosis Journal, and Neurology. Is on the board of directors of Queen Square Analytics. R.O. has received research support from Avid Radiopharmaceuticals, has given lectures in symposia sponsored by GE Healthcare and is an editorial board member of Alzheimer's Research & Therapy and the European Journal of Nuclear Medicine and Molecular Imaging. T.G.B. is a consultant for Aprinoia Therapeutics, Biogen and Avid Radiopharmaceuticals. The remaining authors declare no competing interests.
