## [Transparent Peer Review file · Nature Communications]

Two-step detection of Lewy body pathology via smell-function testing and CSF α -synuclein seed amplification

Corresponding Author: Ms Sophie Mastenbroek

Version 0:

Reviewer comments:

Reviewer #1

(Remarks to the Author)

Mastenbroek et al submit a manuscript detailing a two step approach to identifying likely lewy body pathology by utilizing the UPSIT testing followed by CSF aSyn-SAA testing. The premise is to identify 'high risk' individuals using a non-invasive, inexpensive test, who may then follow up with the more invasive, expensive, and specific aSyn-SAA testing. This is similar to approaches being utilized in the Alzheimer's field using blood based biomarkers to stratify patients who may benefit from PET scans. The work is extremely timely and fills an important gap in the literature. It is well written and utilizes a rare cohort of individuals with antemortem UPSIT testing and autopsy validation with CSF available.

There are some points of clarification that I think would improve the paper and potentially make it more clinically applicable.

There are a handful of issues surrounding the use of post-mortem CSF for the primary analysis here:

- 1) Could the authors please detail what modifications were made to the aSyn-SAA protocol when handling post mortem CSF? The presumption is that sampling after death using an intraventricular approach could introduce small amounts of brain tissue that would increase the likelihood of detecting aSyn seeds. How did the authors address this possibility
- 2) Did these participants have antemortem CSF drawn and are CSF aSyn-SAA results available for comparison to the post-mortem CSF analyses?
- 3) If there are not antemortem CSF results available for comparison, the authors should spend some time in the discussion addressing the limitations of using post mortem CSF and how results may or may not generalize to antemortem samples.

The primary analysis is grounded on the detection of neocortical Lewy body pathology. From the supplement, it appears that the results are less robust in terms of changing the # of LPs required. Previous publications have identified fairly high sensitivity in detecting McKeith Limbic Stage pathology (similar to Stage III in the USSLB) when compared to amygdala-predominant disease (similar to stage IIb). This data appears to be in Figure 1b. The authors should likely discuss the literature on the sensitivity of CSF aSyn-SAA at different Lewy body stages in autopsy validated cohorts (Hall et al 2022, Arnold et al 2022, Tosun et al from ADNI, Bentivenga et al 2024, Samudra et al). This is one of the larger assembled autopsy validated cohorts and is very valuable in that sense.

The ULBSS is one staging system and there are several for Lewy pathology and different hypotheses of how LB pathology may start and progress. Even some debate about whether some patterns are more consequential than others. I would propose the authors perform a parallel analysis where the outcome is gauged at detecting LB that are neocortical OR brainstem/limbic. The basis being that those ULBSS stages are similar to McKeith Limbic and neocortical stages which are felt to be the most likely to result in a lewy body clinical phenotype.

The logistic regression analysis used UPSIT scores as a main part of their model. Were raw UPSIT scores used? Or was it age and sex adjusted percentages (Brumm et al 2024 is the most up to date normative data I believe)? If it was the former, that is a limitation since the same score signifies different degrees of impairment depending on age or sex. If it was the latter, what are the UPSIT percentages that correspond to these '85, 90, 95%' cut-offs? Knowing this would help in the application of this 2-step approach in the future.

The authors can also discuss the literature detailing mild hyposmia in Alzheimer's disease that may contribute to some of the findings the AD group here. Other neurodegenerative diseases have mild hyposmia which the authors can point (ie

hyposmia is not perfectly specifically related to lewy pathology) and the use of CSF aSyn-SAA to follow up on a hypsomic result would be appropriate as a next step. Furthermore the authors can discuss non-degenerative sources of hyposmia like cigarette smoking, sinusitis, sinus surgeries, TBIs etc. Was historical data on any of these factors available?

Why was NPH CSF used as a negative control? LB pathology can occur in patients with NPH. Were these NPH participants known to be LB negative?

There appears to be a few participants with MSA pathology in the cohort. How did their results fare? Were they sorted into the LB- group? Even though MSA is of course primarily a glial synucleinopathy and not considered part of the proposed NSD-ISS, there can be co-occurring neuronal Lewy bodies in MSA cases and sensitivity of CSF aSyn-SAA for aSyn seeding in MSA seems a bit variable still depending on the assay, (also there are a few reports documenting mild hypsomia in MSA).

Minor

There are a handful of typos throughout the manuscript. 42.6% female is cited in the abstract, but 42.7% is written in the results

Line 252: syndrom -> syndrome

Line 253: amnestic -> amnestic

Reviewer #2

(Remarks to the Author)

This is an interesting study examining the use of olfactory testing and CSF SAA in a large cohort of patients with neuropathologically confirmed LBD, with an emphasis on whether olfactory testing allows a preselection of patients that should undergo lumbarpuncture for CSF SAA.

Several aspects should be addressed more clearly:

The abstract is misleading in some sentences: "The two-step workflow identified brain LBP" implies that Lewy body pathology is detected by olfactory testing and/or CSF SAA which is not the case. Lewy body pathology can only be detected by histopathological analysis. Other methods may predict LBP. The sentence "Implementing a two-step workflow in various clinical scenarios may reduce the need for advanced CSF testing" is difficult to understand. It means that screening patients by olfactory testing is useful to select patients for CSF SAA. This should be reported more clearly.

CSF SAA was performed on post-mortem ventricular CSF. It should be discussed if post-mortem CSF differs from pre-mortem CSF (e.g. because there is always a certain time range between death and obtaining post-mortem CSF). It does not really get clear at the beginning of the manuscript that post-mortem CSF was examined. This should be clearly stated, for example in Figure 1.

The patients were very old at the time-point of olfactory testing. Possible confounders of impaired olfaction should be discussed.

The term "LB+ based on CSF α -syn SAA testing" is misleading. A-syn SAA detects a-syn aggregates, not Lewy bodies (that can only be detected by histopathological analysis).

It would be worthwhile to give more information on the clinical picture of the patients. How many fulfilled diagnostic criteria of PD?

Reviewer #3

(Remarks to the Author)

This manuscript from a great group proposes a two-step approach with a smell test and alpha-synuclein Seed Aggregation Assay in cerebrospinal fluid only in those with hyposmia in the first step to reliably detect Lewy pathology.

Overall this is not a new idea and approach. The number of 358 autopsied subjects with an available UPSIT test, that include a pathologic workup is impressive.

My biggest concern is also, that for this study CSF samples have been collected from ventricles and postmortem; it is known that the blood-brain barrier is one of the first to break down after death and it therefore has to be assumed that the CSF is contaminated with brain tissue, making this proposed test for in-vivo diagnostic not reliable. Other cohorts with CSF collection in living subjects will be better to address this and in fact other cohorts have already done this. The authors do not give the interval between death and autopsy.

The UPSIT alone already showed an accuracy of 80% the Seed Aggregation assay increases it to 95%.

There are already substantial studies showing a high accuracy of the Aggregation assay in CSF, that has not been collected by autopsy and that still have additional smell testings done.

Version 1:

Reviewer comments:

Reviewer #1

(Remarks to the Author)

The authors have done a very good job of responding to my comments and the comments of others.

I have a few minor things to add here:

1. The authors have added sections to justify the use of post-mortem CSF and showing that their use may be comparable to ante-mortem CSF in terms of test characteristics and offer the benefit of being most proximal to neuropathological assessment. Can the authors state if they centrifuged samples or discarded overly bloody samples or those with visible tissue pieces in the samples? Other groups have done this.
2. If logistic regression models were used that included UPSIT scores, age, and sex to predict likelihood of SAA positivity, it would be good to document the full models in the supplement to be able to see how much age and sex contributed to the findings. To a certain extent, it may help with the limitation that UPSIT scores were used instead of the age/sex adjusted percentiles. The authors points about derivations of normative values is taken but using the percentiles would make this approach easier to apply in the clinic.
3. The inclusion of the PPMI cohort is impressive and a large undertaking which I appreciate. The authors though should expressly state that PPMI uses a different SAA assay than the one used in the primary analysis and cite appropriate literature if it exists to support the equivalence of the two assays (Kang et al 2019 was a comparative study like this but the assays were different [or different versions]).

Reviewer #2

(Remarks to the Author)

I have no further comments.

Reviewer #3

(Remarks to the Author)

The authors have extensively added further analyses and discussions in the revised version of their manuscript, that supports the outcome of the study. I understand that the big advantage is the postmortal analysis of the brains, but the major concern, that the CSF samples have been collected postmortally even with a short postmortal interval remains. postmortally collected CSF can not be compared to antemortem CSF for which the SAA method has been established.

Version 2:

Reviewer comments:

Reviewer #1

(Remarks to the Author)

The authors have addressed my comments well. Thank you!

Response to reviewers

We thank the Editor and reviewers for taking the time to review our manuscript and providing constructive feedback. We have revised the manuscript accordingly. Please find below the original comments from reviewers in black and our corresponding responses in blue.

Reviewer #1

Mastenbroek et al submit a manuscript detailing a two step approach to identifying likely lewy body pathology by utilizing the UPSIT testing followed by CSF aSyn-SAA testing. The premise is to identify 'high risk' individuals using a non-invasive, inexpensive test, who may then follow up with the more invasive, expensive, and specific aSyn-SAA testing. This is similar to approaches being utilized in the Alzheimer's field using blood based biomarkers to stratify patients who may benefit from PET scans. The work is extremely timely and fills an important gap in the literature. It is well written and utilizes a rare cohort of individuals with antemortem UPSIT testing and autopsy validation with CSF available.

There are some points of clarification that I think would improve the paper and potentially make it more clinically applicable.

Authors' response:

We thank the reviewer for their interest in our work and their suggested improvements. Below, please find a detailed response to each comment.

There are a handful of issues surrounding the use of post-mortem CSF for the primary analysis here:

1) Could the authors please detail what modifications were made to the aSyn-SAA protocol when handling post mortem CSF? The presumption is that sampling after death using an intraventricular approach could introduce small amounts of brain tissue that would increase the likelihood of detecting aSyn seeds. How did the authors address this possibility

Authors' response:

We cannot exclude that the needle entering the ventricles might have drawn in some white matter tissue from the corpus callosum, as that is where the needle enters the lateral ventricles. However, to our knowledge, there is no evidence of synuclein pathology in the deep cerebral white matter or corpus callosum.

We did not modify the CSF α Syn SAA protocol, as it has previously been shown to perform well with *postmortem* CSF samples (Hall et al., 2022). Furthermore, we used polypropylene syringes and tubes for CSF collection, and all handling procedures complied with standard protocols established by ADNI and PPMI (Shaw et al., 2011; Kang et al., 2016).

2) Did these participants have antemortem CSF drawn and are CSF aSyn-SAA results available for comparison to the post-mortem CSF analyses?

Authors' response:

No, participants in this cohort did not undergo *antemortem* CSF sampling. However, we note that the *postmortem* CSF α Syn SAA results in the current study, as well as in our previously

published work (Hall et al., 2022), are consistent with those reported by others using *antemortem* CSF followed by neuropathological examination at autopsy (e.g. Arnold et al., 2022; Bentivenga et al., 2024).

Notably, one study (Arnold et al., 2022) directly compared the sensitivity and specificity of CSF α Syn SAA using both *antemortem* and *postmortem* samples from the same individuals. They found that overall sensitivity was slightly higher in *postmortem* CSF (80.0% vs. 71.2%), likely because *postmortem* CSF samples were collected closer to the time of autopsy. 82.1% of individuals showed concordant CSF α Syn SAA results between *antemortem* and *postmortem* CSF, while 16.1% changed from negative to positive, and only 1.8% from positive to negative.

These findings suggest that *postmortem* CSF α Syn SAA results may be broadly comparable to those obtained *antemortem*, though further research is needed to fully understand the differences. Accordingly, we have added a paragraph in the Discussion section on the differences between *antemortem* and *postmortem* CSF SAA results; please see our response to comment 3 below.

3) If there are not *antemortem* CSF results available for comparison, the authors should spend some time in the discussion addressing the limitations of using *post mortem* CSF and how results may or may not generalize to *antemortem* samples.

Authors' response:

We agree with the reviewer that the manuscript would benefit from a discussion on potential differences between *antemortem* and *postmortem* CSF samples. We have now included a paragraph addressing these considerations in the Discussion section, pages 10-11:

“... Finally, while the use of postmortem CSF samples offers the advantage of collection at the same point in time as the neuropathological assessment, several potential limitations should be considered. First, it has been speculated that postmortem CSF may be affected by overall instability or protein degradation. However, it should be noted that the current cohort has drawn CSF samples from subjects with a short postmortem interval (PMI; median=3.1 hours). Multiple studies, including our own, have demonstrated that postmortem CSF yields results comparable to antemortem CSF regarding analyses such as Western blot, ELISA, proteomic, and metabolomic methodology³³⁻³⁸. Second, concerns have been raised about the equivalence of ventricular versus lumbar CSF. However, a comprehensive proteomics study comparing ventricular and lumbar CSF from the same individuals observed significant differences in protein levels for only two proteins³⁹. Third, prior studies using autopsy-confirmed antemortem CSF α -syn SAA testing reported sensitivity and specificity values for detecting LBP that are similar to those observed in the current study^{3,19-28}. Notably, one study directly compared the sensitivity and specificity of CSF α -syn SAA testing using both antemortem and postmortem samples from the same individuals, and found overall sensitivity to be slightly higher in postmortem CSF (80% vs. 71.2%), possibly because postmortem CSF was collected closer in time to autopsy than antemortem CSF, while specificity was slightly higher in antemortem CSF (98.1% vs. 88.5%)²². While these findings suggest that postmortem CSF SAA results may generally be comparable to those obtained from antemortem samples, further research is needed to fully understand the differences.”

In addition, we have included new analyses assessing the performance of the two-step workflow in individuals from the PPMI cohort with *in vivo* CSF samples. Briefly, we selected all individuals with available UPSIT scores and CSF α -syn SAA testing, excluding those with

a genetic PD mutation and those who were recruited into the prodromal cohort, as their inclusion was based on hyposmia, to avoid bias introduced by UPSIT-driven recruitment. The demographics of the resulting 1209 individuals is shown in Table S10 (see below).

Table S10. PPMI cohort characteristics at baseline

	Whole cohort (N=1209)
Age, years	65.1 (8.7)
Sex, n female	566 (46.8%)
UPSIT score	23.9 (8.2)
CSF SAA+, n	832 (68.8%)
Primary clinical diagnosis	
Idiopathic PD	674 (55.7%)
Alzheimer's disease	2 (0.2%)
Corticobasal syndrome	1 (0.1%)
Dementia with Lewy bodies	5 (0.4%)
Essential tremor	20 (1.7%)
Juvenile autosomal recessive parkinsonism	1 (0.1%)
Control	470 (38.9%)
Spinocerebellar Ataxia	1 (0.1%)
Other neurological disorder(s)	1 (0.1%)

CSF = cerebrospinal fluid; LBP = Lewy body pathology; PD = Parkinson's disease; SAA = Seed Amplification Assay; UPSIT = University of Pennsylvania Smell Identification Test.

As the gold standard of *postmortem* neuropathology was not available in the PPMI cohort, we applied the UPSIT-, age-, and sex-based logistic regression model that was trained on the full autopsy cohort, to the PPMI cohort, to categorize individuals into low and high risk groups for cortical LBP. We then estimated the performance of the UPSIT-based risk classification for predicting CSF α -syn SAA status, and the reduction of required CSF tests by using the UPSIT as a prescreening measure (see Figure 3 below). The UPSIT-based risk classification could accurately predict CSF SAA status, with an accuracy of 79% (95% CI = 77%-81%), a PPV of 82% (95% CI = 80%-85%), and a NPV of 69% (64%-74%). These findings demonstrate that the two-step workflow also performs well in a clinical *in vivo cohort*.

Figure 3. Performance of the two-step workflow in an in vivo clinical cohort

The two-step workflow applied to an in vivo clinical cohort. (a) Distribution of model-derived probabilities for cortical Lewy body pathology (LBP) based on a logistic regression model including UPSIT scores, age, and sex as predictors, trained on the autopsy dataset. A probability threshold corresponding to 95% sensitivity, derived from the autopsy dataset, was used to classify individuals as low (blue dots) or high (red dots) risk for cortical LBP. (b) Performance metrics of the two-step model for predicting CSF α -syn SAA status, including accuracy, positive predictive value (PPV), and negative predictive value (NPV), with 95% bootstrap confidence intervals. (c) Reduction in the number of CSF tests required using the two-step workflow, with 95% bootstrap confidence intervals.

We have added these additional analyses to the manuscript:

Abstract, page 3:

“... The workflow demonstrated strong generalizability in an independent in vivo clinical cohort ($N=1209$; mean age= 65.1 ± 8.7 years; 46.8% female) where it accurately predicted CSF α -syn SAA status (accuracy=79%) and reduced the number of required CSF tests by 26%.”

Introduction, page 5:

“... To confirm that the use of postmortem CSF did not introduce bias, we further validated the two-step workflow in an independent cohort with in vivo CSF α -syn SAA data ($N=1209$).”

Methods, page 12:

“... Detailed information about the PPMI study (ClinicalTrials.gov, number NCT01141023) can be found on the PPMI website (<https://www.ppmi-info.org>) and in previous publications^{40,41}. We included all individuals with UPSIT scores and CSF α -syn SAA testing, excluding those with a genetic PD mutation and those who were recruited into the prodromal cohort, as their inclusion was based on hyposmia, to avoid bias introduced by UPSIT-driven recruitment ($N=1209$). CSF α -syn SAA testing was described in detail previously^{42,43}.”

Methods, page 16:

“... Finally, we applied the UPSIT-, age-, and sex-based logistic regression model that was trained on the full AZSAND/BBDP cohort, to the PPMI cohort, to categorize individuals into low- and high-risk groups for cortical LBP. We then estimated the performance (accuracy,

PPV, and NPV) of the UPSIT-based risk classification for predicting CSF α -syn SAA status, and the reduction of required CSF tests by using the UPSIT as a prescreening measure.”

Results, page 8:

“... To examine whether the use of postmortem CSF influenced the results, we also validated the two-step workflow in an independent cohort (i.e., the Parkinson’s Progression Markers Initiative [PPMI] cohort) consisting of 1209 individuals with in vivo CSF α -syn SAA and UPSIT available. Mean age was 65.1 ± 8.7 years, 46.8% was female, and 68.8% was CSF α -syn SAA+ (Table S10). The majority had a PD diagnosis (55.7%) or was a healthy control (38.9%) (Table S10). We applied the UPSIT-based risk stratification model from the autopsy cohort to the in vivo cohort, where it yielded robust predictions of CSF α -syn SAA status (accuracy=79%; PPV=82%; NPV=69%), while reducing the number of CSF tests required by 26% (Figure 3).”

The primary analysis is grounded on the detection of neocortical Lewy body pathology. From the supplement, it appears that the results are less robust in terms of changing the # of LPs required. Previous publications have identified fairly high sensitivity in detecting McKeith Limbic Stage pathology (similar to Stage III in the USSLB) when compared to amygdala-predominant disease (similar to stage Ib). This data appears to be in Figure 1b. The authors should likely discuss the literature on the sensitivity of CSF α Syn-SAA at different Lewy body stages in autopsy validated cohorts (Hall et al 2022, Arnold et al 2022, Tosun et al from ADNI, Bentivenga et al 2024, Samudra et al). This is one of the larger assembled autopsy validated cohorts and is very valuable in that sense.

Authors’ response:

We have expanded the Discussion section to include a more detailed overview of the literature on the sensitivity of the CSF α Syn SAA across different stages of Lewy body pathology in autopsy-confirmed cohorts. In line with previous studies, our cohort showed high overall specificity (96.8%) and sensitivity (87.0%), with the highest sensitivity observed in cases with widespread LBP and lower sensitivity in cases with more focal pathology.

Discussion, page 9-10:

“... Our findings support previous autopsy-based studies demonstrating the high accuracy of CSF α -syn SAAs for identifying individuals with LBP^{3,18-28}. In our study, the assay demonstrated excellent overall specificity (96.8%) and sensitivity (87.0%) in predicting autopsy-confirmed LBP. Consistent with previous research, sensitivity was highest in individuals with more advanced pathology, reaching 95.6% in cases with brainstem and limbic involvement and 100% in neocortical LBP. Conversely, sensitivity was lower in individuals with focal (early-stage) disease (i.e., 50.0% olfactory bulb and $\approx 70.0\%$ limbic- or brainstem-predominant). These values align with previous studies reporting sensitivities ranging from 14.3% to 63.6% for brainstem- and amygdala-predominant LBP, and from 90% to 100% for limbic and neocortical LBP^{18,22-26,28}. Accordingly, our proposed two-step diagnostic workflow yielded higher accuracy in predicting cortical or brainstem/limbic or neocortical postmortem LBP status compared to predicting the presence of any LBP. The reduced sensitivity of the assay in early-stage cases has been hypothesized to reflect either a lower pathological burden or the presence of a distinct pathological strain of α -syn that may be less well detected by the assay²⁵. Future research should aim to increase sensitivity in focal LBP to improve early detection and clinical applicability.”

The ULBSS is one staging system and there are several for Lewy pathology and different

hypotheses of how LB pathology may start and progress. Even some debate about whether some patterns are more consequential than others. I would propose the authors perform a parallel analysis where the outcome is gauged at detecting LB that are neocortical OR brainstem/limbic. The basis being that those ULBSS stages are similar to McKeith Limbic and neocortical stages which are felt to be the most likely to result in a lewy body clinical phenotype.

Authors' response: We thank the reviewer for this interesting suggestion. We agree that it is important to assess whether our proposed two-step workflow can effectively detect LBP most likely to result in Lewy body-related clinical phenotypes. Accordingly, we have incorporated the suggested analyses into the manuscript.

Briefly, we used classification as USSLB stage III (Brainstem/Limbic) or stage IV (Neocortical) as the neuropathological reference standard. Based on these criteria, 29.9% of individuals were classified as LBP-positive (hereafter referred to as LBP_{B/L-N}), compared to 47.5% for LBP_{any+} and 35.2% for LBP_{ctx+}, as shown in Table S1 below.

Table S1. Participant characteristics at baseline

	Whole cohort (N=358)
Age at death, years	86.2 (7.8)
Sex, n female	153 (42.7%)
UPSIT score	20.8 (8.6)
Interval UPSIT and death, years	3.2 (2.3)
CSF SAA+, n	152 (42.5%)
LBP _{any+} , n	170 (47.5%)
LBP_{B/L-N} , n	126 (35.2%)
LBP _{diffuse+} , n	107 (29.9%)
PMI, hours	3.9 (3.9)

B/L-N = brainstem/limbic or neocortical; CSF = cerebrospinal fluid; ctx = cortex; LBP = Lewy body pathology; PMI = postmortem interval; UPSIT = University of Pennsylvania Smell Identification Test.

In agreement with the lower rate of LBP_{B/L-N} -positivity, more individuals were classified as being at a low risk of having LBP pathology when using LBP_{B/L-N} as the reference standard, compared to LBP_{ctx} and LBP_{any} (Table S7, see below).

In addition, using LBP_{B/L-N} instead of LBP_{ctx} as the reference standard resulted in similar overall accuracies, lower PPVs (particularly among clinically unimpaired individuals where only three participants were LBP_{B/L-N}+), and higher NPVs across all clinical subgroups (Table S8, see below). Slightly more CSF tests were saved when applying the two-step workflow using LBP_{B/L-N} as the reference (48% LBP_{B/L-N} vs LBP_{ctx} 43%) (Figure S3, see below).

These results have been added to the Supplementary (Table S7, Table S8, Figure S1, and Figure S3) and the Results section, pages 7-8:

“... Using postmortem LBP_{any} as the reference standard (instead of LBP_{ctx}) yielded slightly lower accuracies and NPVs, but higher PPVs (Tables S5-S6). A lower reduction in CSF tests was observed (4.6%-38.6%) (Figure S3A). Using $LBP_{B/L-N}$ as the reference standard yielded comparable accuracies, higher NPVs, and lower PPVs than LBP_{ctx} (Tables S7-S8) and slightly more CSF tests were saved (26.7%-84.1%) (Figure S3B). Limiting analyses to individuals who underwent UPSIT testing within 5 years of death did not affect model performance (Table S9).”

Table S7. Model-based risk stratification for brainstem/limbic or neocortical LBP-status

	Whole cohort			Clinical parkinsonism			Clinical AD			Clinically unimpaired		
	All	LBP _{B/L-N-}	LBP _{B/L-N+}	All	LBP _{B/L-N-}	LBP _{B/L-N+}	All	LBP _{B/L-N-}	LBP _{B/L-N+}	All	LBP _{B/L-N-}	LBP _{B/L-N+}
80% sensitivity												
Low risk	227	204 (89.9)	23 (10.1)	65	52 (80.0)	13 (20.0)	58	54 (93.1)	4 (6.9)	42	39 (92.9)	3 (7.1)
High risk	131	47 (35.9)	84 (64.1)	85	13 (15.3)	72 (84.7)	39	21 (53.8)	18 (46.2)	2	2 (100.0)	0 (0.0)
85% sensitivity												
Low risk	213	197 (93.0)	16 (7.0)	56	50 (89.3)	6 (10.7)	53	49 (92.5)	4 (7.5)	42	39 (92.9)	3 (7.1)
High risk	145	54 (37.2)	91 (62.8)	94	15 (16.0)	79 (84.0)	44	26 (59.1)	18 (40.9)	2	2 (100.0)	0 (0.0)
90% sensitivity												
Low risk	199	187 (94.0)	12 (6.0)	50	47 (94.0)	3 (6.0)	49	45 (91.8)	4 (8.2)	41	38 (92.7)	3 (7.3)
High risk	159	64 (40.3)	95 (59.7)	100	18 (18.0)	82 (82.0)	48	30 (62.5)	18 (37.5)	3	3 (100.0)	0 (0.0)
95% sensitivity												
Low risk	171	166 (97.1)	5 (2.9)	40	39 (97.5)	1 (2.5)	40	39 (97.5)	1 (2.5)	37	35 (94.6)	2 (5.4)
High risk	187	85 (45.5)	102 (44.5)	110	26 (23.6)	84 (76.4)	57	36 (63.2)	21 (36.8)	7	6 (85.7)	1 (14.3)

Data are presented as n or n (%). The first column indicates the evaluated strategies with different sensitivity-based thresholds for UPSIT-derived risk stratification. For each strategy, the total number of individuals in the low- and high-risk groups are shown, followed by numbers of Lewy body pathology negative (LBP-) and LBP+ participants according to postmortem neuropathology measures corresponding to Unified Staging System for Lewy Body Disorders (USSLB) stages III. Brainstem/Limbic or IV. Neocortical (B/L-N). The percentage of LBP_{B/L-N-}-negatives in the low-risk group and the percentage of LBP_{B/L-N+}-positives in the high-risk group correspond to each evaluated threshold's NPV and PPV, respectively.

AD = Alzheimer's disease; B/L-N = brainstem/limbic or neocortical; CSF = cerebrospinal fluid; LB = Lewy body

Table S8 Two-step workflow performance for detecting brainstem/limbic or neocortical LBP-status

Approach	Sensitivity	Median accuracy (CI)	Median PPV (CI)	Median NPV (CI)	Median TP (CI)	Median FP (CI)	Median TN (CI)	Median FN (CI)
Whole cohort								
Two-step	80%	0.90 (0.89-0.91)	0.86 (0.84-0.88)	0.92 (0.90-0.93)	85 (82-88)	14 (12-16)	237 (235-239)	22 (19-25)
Two-step	85%	0.91 (0.90-0.91)	0.85 (0.83-0.86)	0.93 (0.92-0.94)	90 (87-93)	16 (15-18)	235 (233-236)	17 (14-20)
Two-step	90%	0.91 (0.91-0.92)	0.84 (0.83-0.85)	0.95 (0.94-0.96)	95 (93-97)	19 (17-20)	232 (231-234)	12 (10-14)
Two-step	95%	0.92 (0.91-0.92)	0.81 (0.80-0.82)	0.97 (0.97-0.98)	101 (99-102)	24 (22-26)	227 (225-229)	6 (5-8)
UPSIT-only	80%	0.81 (0.80-0.82)	0.65 (0.63-0.64)	0.90 (0.89-0.91)	85 (82-88)	46 (43-49)	205 (202-208)	22 (19-25)
UPSIT-only	85%	0.80 (0.79-0.82)	0.63 (0.62-0.64)	0.92 (0.91-0.93)	90 (87-93)	53 (50-56)	198 (195-201)	17 (14-20)
UPSIT-only	90%	0.79 (0.78-0.80)	0.60 (0.58-0.62)	0.94 (0.93-0.95)	96 (93-98)	64 (60-68)	187 (183-191)	11 (9-14)
UPSIT-only	95%	0.74 (0.73-0.75)	0.54 (0.53-0.55)	0.97 (0.96-0.98)	102 (100-103)	87 (82-91)	165 (160-169)	5 (4-7)
CSF-only	-	0.86 (0.83-0.89)	0.69 (0.62-0.76)	0.99 (0.98-1.00)	105 (88-122)	47 (35-59)	204 (186-222)	2 (0-5)
Clinical parkinsonism								
Two-step	80%	0.90 (0.88-0.92)	0.95 (0.95-0.96)	0.85 (0.81-0.88)	74 (71-77)	4 (3-4)	61 (61-62)	11 (8-14)
Two-step	85%	0.93 (0.91-0.94)	0.95 (0.94-0.95)	0.91 (0.87-0.94)	79 (76-81)	4 (4-5)	61 (60-61)	6 (4-9)
Two-step	90%	0.94 (0.93-0.95)	0.93 (0.92-0.94)	0.95 (0.92-0.95)	82 (80-82)	6 (5-7)	59 (58-60)	3 (3-5)
Two-step	95%	0.93 (0.91-0.94)	0.90 (0.88-0.91)	0.97 (0.95-0.98)	83 (82-84)	9 (8-11)	56 (54-57)	2 (1-3)
UPSIT-only	80%	0.84 (0.81-0.86)	0.85 (0.84-0.87)	0.83 (0.79-0.86)	74 (71-77)	13 (11-14)	52 (51-54)	11 (8-14)
UPSIT-only	85%	0.86 (0.84-0.87)	0.84 (0.83-0.85)	0.89 (0.85-0.93)	79 (76-81)	15 (14-16)	50 (49-51)	6 (4-9)
UPSIT-only	90%	0.86 (0.84-0.87)	0.82 (0.80-0.84)	0.94 (0.90-0.96)	82 (80-83)	18 (16-20)	47 (45-49)	3 (2-5)
UPSIT-only	95%	0.83 (0.81-0.84)	0.77 (0.75-0.78)	0.98 (0.95-1.00)	84 (83-85)	26 (4-28)	39 (37-41)	1 (0-2)
CSF-only	-	0.91 (0.86-0.95)	0.87 (0.80-0.93)	0.98 (0.94-1.00)	84 (72-96)	13 (7-19)	52 (41-64)	1 (0-3)
Clinical Alzheimer's disease								
Two-step	80%	0.90 (0.88-0.91)	0.77 (0.73-0.81)	0.93 (0.92-0.95)	17 (16-18)	5 (4-6)	70 (69-71)	5 (4-6)
Two-step	85%	0.89 (0.88-0.91)	0.74 (0.69-0.78)	0.94 (0.93-0.95)	18 (17-18)	6 (5-8)	69 (67-70)	4 (4-5)
Two-step	90%	0.89 (0.88-0.91)	0.70 (0.69-0.74)	0.96 (0.94-0.97)	19 (18-20)	8 (7-8)	67 (67-68)	3 (2-4)

Two-step	95%	0.90 (0.89-0.91)	0.70 (0.69-0.72)	0.99 (0.97-0.99)	21 (20-21)	9 (8-9)	66 (66-67)	1 (1-2)
UPSIT-only	80%	0.74 (0.71-0.76)	0.46 (0.43-0.49)	0.92 (0.90-0.93)	17 (16-18)	20 (18-23)	55 (52-57)	5 (4-6)
UPSIT-only	85%	0.70 (0.68-0.73)	0.42 (0.40-0.45)	0.93 (0.91-0.93)	18 (17-18)	24 (22-27)	51 (48-53)	4 (4-5)
UPSIT-only	90%	0.66 (0.64-0.69)	0.39 (0.37-0.41)	0.94 (0.92-0.96)	19 (18-20)	29 (27-31)	46 (44-48)	3 (2-4)
UPSIT-only	95%	0.61 (0.59-0.63)	0.36 (0.35-0.38)	0.97 (0.95-0.98)	21 (20-21)	37 (34-39)	38 (36-41)	1 (1-2)
CSF-only	-	0.86 (0.78-0.93)	0.61 (0.46-0.79)	0.99 (0.95-1.00)	21 (13-29)	13 (6-19)	62 (52-71)	1 (0-3)
Clinically unimpaired								
Two-step	80%	0.91 (0.91-0.91)	0.00 (0.00-0.00)	0.93 (0.93-0.93)	0 (0-0)	1 (1-1)	40 (40-40)	3 (3-3)
Two-step	85%	0.91 (0.91-0.91)	0.00 (0.00-0.00)	0.93 (0.93-0.93)	0 (0-0)	1 (1-1)	40 (40-40)	3 (3-3)
Two-step	90%	0.91 (0.91-0.91)	0.00 (0.00-0.00)	0.93 (0.93-0.93)	0 (0-0)	1 (1-1)	40 (40-40)	3 (3-3)
Two-step	95%	0.89 (0.89-0.93)	0.00 (0.00-0.50)	0.93 (0.93-0.98)	0 (0-2)	2 (1-2)	39 (39-40)	3 (1-3)
UPSIT-only	80%	0.89 (0.89-0.89)	0.00 (0.00-0.00)	0.93 (0.93-0.93)	0 (0-0)	2 (2-2)	39 (39-39)	3 (3-3)
UPSIT-only	85%	0.89 (0.89-0.89)	0.00 (0.00-0.00)	0.93 (0.93-0.93)	0 (0-0)	2 (2-2)	39 (39-39)	3 (3-3)
UPSIT-only	90%	0.86 (0.84-0.89)	0.00 (0.00-0.00)	0.93 (0.93-0.93)	0 (0-0)	3 (2-4)	38 (37-39)	3 (3-3)
UPSIT-only	95%	0.80 (0.77-0.84)	0.00 (0.00-0.00)	0.92 (0.92-0.97)	0 (0-2)	7 (5-7)	34 (34-36)	3 (1-3)
CSF-only	-	0.82 (0.71-0.93)	0.00 (0.00-1.00)	1.00 (1.00-1.00)	3 (0-7)	8 (3-13)	33 (27-38)	0 (0-0)

Data correspond to the model across 1000 iterations, yielding median accuracy. The 95% confidence interval is shown across 1000 iterations. FN = false negative; FP = false positive; NPV = negative predictive value PPV = positive predictive value; TN = true negative; TP = true positive.

Figure S3. Results summary against any and brainstem/limbic or neocortical LBP status

a

Patient population	Smell test-based high risk (95% sensitivity)	Lewy body positivity based on CSF SAA in high risk group	Lewy body positivity based on postmortem neuropathology in high risk CSF SAA positive group	Reduction in CSF tests
Whole cohort, n = 358, 47.5% LBP _{any+}	→ 85.5%	→ 48.0%	→ 95.9%	→ 14.5%
Parkinsonian symptoms, n = 150, 70.7% LBP _{any+}	→ 95.3%	→ 67.8%	→ 99.0%	→ 4.7%
Clinical Alzheimer's disease, n = 97, 44.3% LBP _{any+}	→ 88.7%	→ 39.5%	→ 97.1%	→ 11.3%
Clinically unimpaired n = 44, 22.7% LBP _{any+}	→ 61.4%	→ 25.9%	→ 85.7%	→ 38.6%

b

Patient population	Smell test-based high risk (95% sensitivity)	Lewy body positivity based on CSF SAA in high risk group	Lewy body positivity based on postmortem neuropathology in high risk CSF SAA positive group	Reduction in CSF tests
Whole cohort, n = 358, 29.9% LBP _{B/L-N+}	→ 52.2%	→ 48.0%	→ 66.8%	→ 47.8%
Parkinsonian symptoms, n = 150, 56.7% LBP _{B/L-N+}	→ 73.3%	→ 67.8%	→ 83.6%	→ 26.7%
Clinical Alzheimer's disease, n = 97, 22.7% LBP _{B/L-N+}	→ 58.8%	→ 39.5%	→ 52.6%	→ 41.2%
Clinically unimpaired n = 44, 6.8% LBP _{B/L-N+}	→ 15.9%	→ 25.9%	→ 42.9%	→ 84.1%

Summary of the proportion of individuals selected as high-risk, Lewy body (LB)-positive based on CSF, LB-positive based on postmortem neuropathology in (a) any region (LBP_{any+}) and (b) brainstem and limbic or neocortical LBP stages (LBP_{B/L-N}) and the reduction in CSF tests, for each of the four clinical subgroups.

B/L-N = brainstem/limbic or neocortical; CSF = cerebrospinal fluid; LBP = Lewy body pathology; SAA = seed amplification assay.

The logistic regression analysis used UPSIT scores as a main part of their model. Were raw UPSIT scores used? Or was it age and sex adjusted percentages (Brumm et al 2024 is the most up to date normative data I believe)? If it was the former, that is a limitation since the same score signifies different degrees of impairment depending on age or sex. If it was the latter, what are the UPSIT percentages that correspond to these '85, 90, 95%' cut-offs? Knowing this would help in the application of this 2-step approach in the future.

Authors' response:

Rather than using age- and sex-specific normative UPSIT-scores, we chose to include raw UPSIT scores, age, and sex as separate predictors in the logistic regression analysis. This approach allows the model to account for the fact that individuals with the same UPSIT score may have different probabilities of having neuropathologically confirmed LBP, depending on their sex and age (i.e., male sex and older age are associated with a higher probability). We opted for this method because normative scores are often derived from clinically defined control groups, rather than autopsy-confirmed control groups, which may include individuals with undetected LBP or other brain pathologies.

We have added our rationale to the Methods section, page 15:

"... We chose to add age and sex as independent predictors, rather than using normative scores, as such scores are often derived from clinically defined control groups, which may include individuals with undetected LBP or other brain pathologies."

The authors can also discuss the literature detailing mild hyposmia in Alzheimer's disease that may contribute to some of the findings the AD group here. Other neurodegenerative diseases

have mild hyposmia which the authors can point (ie hyposmia is not perfectly specifically related to lewy pathology) and the use of CSF aSyn-SAA to follow up on a hypsomic result would be appropriate as a next step. Furthermore the authors can discuss non-degenerative sources of hyposmia like cigarette smoking, sinusitis, sinus surgeries, TBIs etc. Was historical data on any of these factors available?

Authors' response:

Thank you for this thoughtful comment. We have now added information on the presence of non-neurodegenerative factors (i.e., smoking, alcohol consumption, history of traumatic brain injury, and history of sinusitis or rhinitis) to the Supplementary Results (Table S11). We compared the presence of such factors between individuals classified as high risk for LBP based on their UPSIT score, but negative for *postmortem* LBP_{ctx} (i.e., false positives) and true positives (i.e., high risk and LBP_{ctx}⁺). No statistically significant differences were observed, suggesting that in the current study, non-neurodegenerative factors did not seem to significantly influence UPSIT-based risk classification for LBP.

Table S11. Non-degenerative confounders of olfactory function in false positives and true positives based on UPSIT risk-stratification

	False positive (N=107)	True positive (N=131)	P value
Smoking (>0 cigarettes a day)			0.188
No	44 (41.1%)	76 (58.0%)	
Yes	63 (58.9%)	55 (42.0%)	
Alcohol (>0 units per day)			0.173
No	75 (70.1%)	107 (81.7%)	
Yes	32 (29.9%)	24 (18.3%)	
History of traumatic brain injury			0.620
No	83 (77.6%)	104 (79.4%)	
Yes	24 (22.4%)	27 (20.6%)	
History of sinusitis / rhinitis			0.176
No	78 (72.9%)	98 (74.8%)	
Yes	29 (27.1%)	33 (25.2%)	

Presence of potential confounders of olfactory impairment. P value indicates results from chi-squared tests.

In addition, we have added a paragraph in the Discussion section noting that other neurodegenerative and non-neurodegenerative factors can influence olfactory function, page 10:

*“... It should be noted that olfactory impairment is not specific to Lewy body disease and has been observed across a range of other neurodegenerative diseases, including AD, multiple sclerosis (MS), amyotrophic lateral sclerosis (ALS), and Huntington’s disease (HD)²⁹⁻³². In addition, olfactory function can be affected by several non-degenerative factors, as the nasal neuroepithelium is in direct communication with the external environment. For instance, smoking, head trauma, and chronic sinonasal diseases have all been associated with reduced olfactory function (see **Table S11** for the frequencies of these factors in our cohort)³². Factors influencing smell function might confound the interpretation of the UPSIT results in the context of Lewy body disease, potentially leading to false positives, but not false negatives. In our*

proposed two-step workflow, the inclusion of the CSF α -syn SAA test as a confirmatory measure might help mitigate this limitation, as shown by the increased accuracy of the two-step approach compared to using the UPSIT alone.”

Why was NPH CSF used as a negative control? LB pathology can occur in patients with NPH. Were these NPH participants known to be LB negative?

Authors’ response:

NPH participants provide large volumes of samples, which can be used several times in different SAA runs. All NPH samples that we used as negative controls have been run and confirmed as negative at least three times.

There appears to be a few participants with MSA pathology in the cohort. How did their results fare? Were they sorted into the LB- group? Even though MSA is of course primarily a glial synucleinopathy and not considered part of the proposed NSD-ISS, there can be co-occurring neuronal Lewy bodies in MSA cases and sensitivity of CSF aSyn-SAA for aSyn seeding in MSA seems a bit variable still depending on the assay, (also there are a few reports documenting mild hypsomnia in MSA).

Authors’ response:

We thank the reviewer for this interesting comment. Three individuals in our cohort received a clinicopathological diagnosis of MSA at autopsy. None of these showed any Lewy body neuropathology in any of the 10 tested brain regions. However, one individual did test positive on the CSF aSyn SAA. This individual was a 70-year-old male with an UPSIT score of 31 out of 40. Using LBP_{ctx} as the reference standard, this individual was classified as low risk of LBP (probability = 15.1%), and would not have been CSF tested based on the two-step workflow.

We have added this result to the Results section, page 6:

“... Among three individuals with multiple system atrophy (MSA) (Table S2), none showed evidence of postmortem LBP in any of the 10 tested brain regions, while one individual tested positive on the CSF α -syn SAA.”

Minor

There are a handful of typos throughout the manuscript. 42.6% female is cited in the abstract, but 42.7% is written in the results

Line 252: syndrom -> syndrome

Line 253: amnestic -> amnestic

Authors’ response:

We thank the reviewer for pointing this out. We have thoroughly reread the manuscript and corrected all identified errors.

Reviewer #2:

This is an interesting study examining the use of olfactory testing and CSF SAA in a large cohort of patients with neuropathologically confirmed LBD, with an emphasis on whether olfactory testing allows a preselection of patients that should undergo lumbarpuncture for CSF SAA.

Authors' response:

We thank the reviewer for their interest in our work and their suggested improvements. Please find a detailed response to each comment below.

Several aspects should be addressed more clearly:

The abstract is misleading in some sentences: "The two-step workflow identified brain LBP" implies that Lewy body pathology is detected by olfactory testing and/or CSF SAA which is not the case. Lewy body pathology can only be detected by histopathological analysis. Other methods may predict LBP.

Authors' response:

We have changed the wording "identifying" to "predicting" LBP throughout the manuscript when discussing olfactory testing and CSF SAA analyses.

The sentence "Implementing a two-step workflow in various clinical scenarios may reduce the need for advanced CSF testing" is difficult to understand. It means that screening patients by olfactory testing is useful to select patients for CSF SAA. This should be reported more clearly.

Authors' response:

We have formulated the implications of a two-step workflow more clearly in the Abstract, page 3:

"... Implementing a two-step workflow in which patients first undergo a prescreening using olfactory function testing, can help identify those who are most suitable for confirmatory CSF testing. This approach may reduce the total number of necessary CSF tests, alleviating the burden on tested individuals and lowering healthcare costs."

CSF SAA was performed on post-mortem ventricular CSF. It should be discussed if post-mortem CSF differs from pre-mortem CSF (e.g. because there is always a certain time range between death and obtaining post-mortem CSF). It does not really get clear at the beginning of the manuscript that post-mortem CSF was examined. This should be clearly stated, for example in Figure 1.

Authors' response:

We have clarified throughout the manuscript and Figure 1 that the current study uses *postmortem* CSF.

e.g., abstract, page 3:

"... A total of 358 autopsied participants with smell testing, postmortem CSF sampling, and neuropathological LBP assessment were included..."

e.g., introduction, page 4, third paragraph:

"... This was performed in a heterogenous longitudinal cohort (N=358) with antemortem UPSIT scores, postmortem CSF α -syn SAA results, and postmortem neuropathological assessments of regional LBP load."

e.g., Figure 1, panel C and legend:

Figure 1. Two-step workflow design and results summary

a Design of a conditional two-step workflow to detect Lewy body pathology according to CSF α syn SAA testing

b Association between *postmortem* Lewy body pathology, CSF α syn SAA testing, and *antemortem* smell function

c Summary of the proportion of individuals at high risk of having *postmortem* Lewy body pathology in the cortex

Patient population	Small test-based high risk (95% sensitivity)	Lewy body positivity based on post mortem CSF SAA in high risk group	Lewy body positivity based on postmortem neuropathology in high risk CSF SAA positive group	Reduction in CSF tests
Whole cohort, $n = 358$, 35.2% LBP _{ctx} +	57.0%	64.2%	90.1%	43%
Parkinsonian symptoms, $n = 150$, 62.7% LBP _{ctx} +	77.7%	81.7%	96.8%	23%
Clinical Alzheimer's disease, $n = 97$, 26.8% LBP _{ctx} +	66.0%	48.4%	80.6%	35%
Clinically unimpaired $n = 44$, 11.4% LBP _{ctx} +	20.5%	44.4%	75.0%	80%

(a) Design of a two-step workflow to detect Lewy body pathology (LBP). Step 1 consists of UPSIT-based risk stratification into high- and low-risk groups having cortical LBP (LBP_{ctx}). Step 2 includes confirmatory CSF α -syn seeding amplification assay (SAA) testing in high-risk patients identified in step 1. (b) The association between *postmortem* LBP, *postmortem* CSF α -syn SAA, and UPSIT scores. (c) Summary of the proportion of individuals selected as high-risk, LB-positive based on *postmortem* CSF, LB-positive based on *postmortem* neuropathology and the reduction in CSF tests, for each of the four scenarios.

In addition, we have added a paragraph to the Discussion section in which we extensively discuss literature on the comparison between *antemortem* and *postmortem* CSF (pages 10-11):

“... Finally, while the use of *postmortem* CSF samples offers the advantage of collection at the same point in time as the neuropathological assessment, several potential limitations should be considered. First, it has been speculated that *postmortem* CSF may be affected by overall instability or protein degradation. However, it should be noted that the current cohort has drawn CSF samples from subjects with a short *postmortem* interval (PMI; median=3.1 hours). Multiple studies, including our own, have demonstrated that *postmortem* CSF yields results comparable to *antemortem* CSF regarding analyses such as Western blot, ELISA, proteomic, and metabolomic methodology³³⁻³⁸. Second, concerns have been raised about the equivalence of ventricular versus lumbar CSF. However, a comprehensive proteomics study comparing ventricular and lumbar CSF from the same individuals observed significant differences in

protein levels for only two proteins³⁹. Third, prior studies using autopsy-confirmed antemortem CSF α -syn SAA testing reported sensitivity and specificity values for detecting LBP that are similar to those observed in the current study^{3,19-28}. Notably, one study directly compared the sensitivity and specificity of CSF α -syn SAA testing using both antemortem and postmortem samples from the same individuals, and found overall sensitivity to be slightly higher in postmortem CSF (80% vs. 71.2%), possibly because postmortem CSF was collected closer in time to autopsy than antemortem CSF, while specificity was slightly higher in antemortem CSF (98.1% vs. 88.5%)²². While these findings suggest that postmortem CSF SAA results may generally be comparable to those obtained from antemortem samples, further research is needed to fully understand the differences.”

In addition, we have included new analyses assessing the performance of the two-step workflow in individuals from the PPMI cohort with *in vivo* CSF samples. Briefly, we selected all individuals with available UPSIT scores and CSF α -syn SAA testing, excluding those with a genetic PD mutation and those who were recruited into the prodromal cohort, as their inclusion was based on hyposmia, to avoid bias introduced by UPSIT-driven recruitment. The demographics of the resulting 1209 individuals is shown in Table S10 (see below).

Table S10. PPMI cohort characteristics at baseline

	Whole cohort (N=1209)
Age, years	65.1 (8.7)
Sex, n female	566 (46.8%)
UPSIT score	23.9 (8.2)
CSF SAA+, n	832 (68.8%)
Primary clinical diagnosis	
Idiopathic PD	674 (55.7%)
Alzheimer’s disease	2 (0.2%)
Corticobasal syndrome	1 (0.1%)
Dementia with Lewy bodies	5 (0.4%)
Essential tremor	20 (1.7%)
Juvenile autosomal recessive parkinsonsim	1 (0.1%)
Control	470 (38.9%)
Spinocerebellar Ataxia	1 (0.1%)
Other neurological disorder(s)	1 (0.1%)

CSF = cerebrospinal fluid; LBP = Lewy body pathology; PD = Parkinson’s disease; SAA = Seed Amplification Assay; UPSIT = University of Pennsylvania Smell Identification Test.

As the gold standard of *postmortem* neuropathology was not available in the PPMI cohort, we applied the UPSIT-, age-, and sex-based logistic regression model that was trained on the full autopsy cohort, to the PPMI cohort, to categorize individuals into low and high risk groups for

cortical LBP. We then estimated the performance of the UPSIT-based risk classification for predicting CSF α -syn SAA status, and the reduction of required CSF tests by using the UPSIT as a prescreening measure (see Figure 3 below). The UPSIT-based risk classification could accurately predict CSF SAA status, with an accuracy of 79% (95% CI = 77%-81%), a PPV of 82% (95% CI = 80%-85%), and a NPV of 69% (64%-74%). These findings demonstrate that the two-step workflow also performs well in a clinical *in vivo* cohort.

Figure 3. Performance of the two-step workflow in an *in vivo* clinical cohort

The two-step workflow applied to an *in vivo* clinical cohort. (a) Distribution of model-derived probabilities for cortical Lewy body pathology (LBP) based on a logistic regression model including UPSIT scores, age, and sex as predictors, trained on the autopsy dataset. A probability threshold corresponding to 95% sensitivity, derived from the autopsy dataset, was used to classify individuals as low (blue dots) or high (red dots) risk for cortical LBP. (b) Performance metrics of the two-step model for predicting CSF α -syn SAA status, including accuracy, positive predictive value (PPV), and negative predictive value (NPV), with 95% bootstrap confidence intervals. (c) Reduction in the number of CSF tests required using the two-step workflow, with 95% bootstrap confidence intervals.

We have added these additional analyses to the manuscript:

Abstract, page 3:

“... The workflow demonstrated strong generalizability in an independent *in vivo* clinical cohort ($N=1209$; mean age= 65.1 ± 8.7 years; 46.8% female) where it accurately predicted CSF α -syn SAA status (accuracy=79%) and reduced the number of required CSF tests by 26%.”

Introduction, page 5:

“... To confirm that the use of postmortem CSF did not introduce bias, we further validated the two-step workflow in an independent cohort with *in vivo* CSF α -syn SAA data ($N=1209$).”

Methods, page 12:

“... Detailed information about the PPMI study (ClinicalTrials.gov, number NCT01141023) can be found on the PPMI website (<https://www.ppmi-info.org>) and in previous publications^{40,41}. We included all individuals with UPSIT scores and CSF α -syn SAA testing, excluding those with a genetic PD mutation and those who were recruited into the prodromal

cohort, as their inclusion was based on hyposmia, to avoid bias introduced by UPSIT-driven recruitment (N=1209). CSF α -syn SAA testing was described in detail previously^{42,43}.

Methods, page 16:

“... Finally, we applied the UPSIT-, age-, and sex-based logistic regression model that was trained on the full AZSAND/BBDP cohort, to the PPMI cohort, to categorize individuals into low- and high-risk groups for cortical LBP. We then estimated the performance (accuracy, PPV, and NPV) of the UPSIT-based risk classification for predicting CSF α -syn SAA status, and the reduction of required CSF tests by using the UPSIT as a prescreening measure.”

Results, page 8:

*“... To examine whether the use of postmortem CSF influenced the results, we also validated the two-step workflow in an independent cohort (i.e., the Parkinson’s Progression Markers Initiative [PPMI] cohort) consisting of 1209 individuals with in vivo CSF α -syn SAA and UPSIT available. Mean age was 65.1+8.7 years, 46.8% was female, and 68.8% was CSF α -syn SAA+ (**Table S10**). The majority had a PD diagnosis (55.7%) or was a healthy control (38.9%) (**Table S10**). We applied the UPSIT-based risk stratification model from the autopsy cohort to the in vivo cohort, where it yielded robust predictions of CSF α -syn SAA status (accuracy=79%; PPV=82%; NPV=69%), while reducing the number of CSF tests required by 26% (**Figure 3**).”*

The patients were very old at the time-point of olfactory testing. Possible confounders of impaired olfaction should be discussed.

Authors’ response:

We have added a paragraph in the Discussion section noting that other neurodegenerative and non-neurodegenerative factors can influence olfactory function, page 10:

*“... It should be noted that olfactory impairment is not specific to Lewy body disease and has been observed in a range of other neurodegenerative diseases, including AD, multiple sclerosis (MS), amyotrophic lateral sclerosis (ALS), Huntington’s disease (HD), and Multiple system atrophy (MSA)²⁹⁻³². In addition, olfactory function can be affected by several non-degenerative factors, as the nasal neuroepithelium is in direct communication with the external environment. For instance, smoking, head trauma, and chronic sinonasal diseases have all been associated with reduced olfactory function (see **Table S11** for the frequencies of these factors in our cohort)³². Factors influencing smell function might confound the interpretation of the UPSIT results in the context of Lewy body disease. However, in our proposed two-step workflow, the inclusion of the CSF α -syn SAA test as a confirmatory measure might help mitigate this limitation, as shown by the increased accuracy of the two-step approach compared to using the UPSIT alone.”*

The term "LB+ based on CSF α -syn SAA testing" is misleading. A-syn SAA detects a-syn aggregates, not Lewy bodies (that can only be detected by histopathological analysis).

Authors’ response:

We have changed the term “LB+ based on CSF testing” to “CSF α -syn SAA-positive” throughout the manuscript.

e.g., results section, page 6, second paragraph:

“...The proportion of individuals classified as CSF α -syn SAA+ increased with more widespread postmortem LBP...”

“...UPSIT scores decreased with more advanced postmortem LBP. Individuals classified as LBP+ based on neuropathology (LBP_{ctx}+, LBP_{any}+, or LBP_{B/L-N}+) and individuals with a positive CSF α -syn SAA result had lower UPSIT scores, reflecting worse olfactory function...”

It would be worthwhile to give more information on the clinical picture of the patients. How many fulfilled diagnostic criteria of PD?

Authors’ response:

Clinical diagnostic classification was performed after each annual assessment at a consensus conference attended by neurologists, psychiatrists, and neuropsychologists. An overview of the clinical diagnoses in the present study are shown in Table S3 (see below). 17.9% fulfilled criteria for PD, 1.1% probable PD, and 0.6% possible PD. 2.0% received a DLB diagnosis. We have added this information to the Supplementary.

Table S3. Antemortem clinical diagnosis

Clinical diagnosis	Overall (N=358)
AD	3 (0.8%)
AD and Lewy body disease	1 (0.3%)
AD and/or CvD	1 (0.3%)
AD and DLB	1 (0.3%)
AD and VaD	6 (1.7%)
ALS	1 (0.3%)
Clinically impaired, NOS	86 (24.0%)
Clinically impaired, NOS or FTD	1 (0.3%)
DLB	7 (2.0%)
DLB, AD, or PD	1 (0.3%)
DLB or probable AD	1 (0.3%)
Cognitively unimpaired	78 (21.8%)
Cognitively unimpaired or clinically impaired NOS	4 (1.1%)
Cognitively unimpaired, parkinsonism	5 (1.4%)
Mixed AD	1 (0.3%)
Mixed VaD	2 (0.6%)
MSA	1 (0.3%)
MSA and NPH	1 (0.3%)
PD	64 (17.9%)
PD and DLB	1 (0.3%)
PD or DLB	1 (0.3%)
PD, DLB, FTD, or PSP	1 (0.3%)
PD, MSA, PSP, or CBD	1 (0.3%)
PLS	1 (0.3%)
Possible AD	18 (5.0%)
Possible AD and VaD	1 (0.3%)
Possible AD or VaD	1 (0.3%)
Possible DLB	1 (0.3%)

Possible DLB, CBD, or MSA	1 (0.3%)
Possible PD	2 (0.6%)
PPA due to FTD or AD	2 (0.6%)
PPA NOS	1 (0.3%)
Probable AD	44 (12.3%)
Probable AD and DLB	2 (0.6%)
Probable PD	4 (1.1%)
PSP	3 (0.8%)
VaD	8 (2.2%)

AD = Alzheimer's disease; ALS = amyotrophic lateral sclerosis; CBD = corticobasal degeneration ;CvD = cerebrovascular disease; DLB = dementia with Lewy bodies; FTD = frontotemporal dementia; MSA = multiple system atrophy; NOS = not otherwise specified; NPH = normal pressure hydrocephalus; PD = Parkinson's disease; PLS = primary lateral sclerosis; PPA = primary progressive aphasia; PSP = progressive supranuclear palsy; VaD = vascular dementia.

Reviewer #3:

This manuscript from a great group proposes a two-step approach with a smell test and alpha-synuclein Seed Aggregation Assay in cerebrospinal fluid only in those with hyposmia in the first step to reliably detect Lewy pathology.

Overall this is not a new idea and approach. The number of 358 autopsied subjects with an available UPSIT test, that include a pathologic workup is impressive.

Authors' response:

We thank the reviewer for their appreciation of our work and their suggested improvements. Below, please find a detailed response to each comment.

My biggest concern is also, that for this study CSF samples have been collected from ventricles and postmortem; it is known that the blood-brain barrier is one of the first to break down after death and it therefore has to be assumed that the CSF is contaminated with brain tissue, making this proposed test for in-vivo diagnostic not reliable. Other cohorts with CSF collection in living subjects will be better to address this and in fact other cohorts have already done this.

Authors' response:

We thank the reviewer for raising this important point regarding the use of *postmortem* CSF. We agree that the differences between *antemortem* and *postmortem* CSF should be acknowledged and carefully considered. However, we argue that using *postmortem* CSF in our study is justified for several reasons.

First, the *postmortem* CSF SAA results in the current study and in our previously published study (Hall et al., 2022) have shown consistency with those reported by others who have used *antemortem* CSF with subsequent neuropathological examination at autopsy (e.g. Arnold et al., 2022; Bentivenga et al., 2024). We have expanded the Discussion section to include a detailed comparison of our results with those from studies using *in vivo* CSF.

Discussion, page 9-10:

"... Our findings support previous autopsy-based studies demonstrating the high accuracy of CSF α -syn SAAs for identifying individuals with LBP^{3,18-28}. In our study, the assay

demonstrated excellent overall specificity (96.8%) and sensitivity (87.0%) in predicting autopsy-confirmed LBP. Consistent with previous research, sensitivity was highest in individuals with more advanced pathology, reaching 95.6% in cases with brainstem and limbic involvement and 100% in neocortical LBP. Conversely, sensitivity was lower in individuals with focal (early-stage) disease (i.e., 50.0% olfactory bulb and ≈70.0% limbic- or brainstem-predominant). These values align with previous studies reporting sensitivities ranging from 14.3% to 63.6% for brainstem- and amygdala-predominant LBP, and from 90% to 100% for limbic and neocortical LBP^{18,22-26,28}. Accordingly, our proposed two-step diagnostic workflow yielded higher accuracy in predicting cortical or diffuse postmortem LBP status compared to predicting the presence of any LBP. The reduced sensitivity of the assay in early-stage cases has been hypothesized to reflect either a lower pathological burden or the presence of a distinct pathological strain of α -syn that may be less well detected by the assay²⁵. Future research should aim to increase sensitivity in focal LBP to improve early detection and clinical applicability.”

Second a recent study (Arnold et al., 2022) directly compared the sensitivity and specificity of the CSF α Syn SAA using both *antemortem* and *postmortem* CSF from the same individuals. This study found slightly higher sensitivity in *postmortem* CSF (80.0% vs. 71.2%), likely due to *postmortem* CSF being collected closer to autopsy. 82.1% of individuals had concordant results between *antemortem* and *postmortem* CSF α Syn SAA status, 16.1% changed from negative to positive, and only 1.8% from positive to negative.

Third, although *postmortem* CSF could theoretically be influenced by instability or protein degradation, the CSF samples in the current cohort were collected from subjects with a short *postmortem* interval (median=3.1 hours), which minimizes these concerns. Furthermore, several studies, including our own, have demonstrated that *postmortem* CSF yields results comparable to *antemortem* CSF in analyses such as Western blot, ELISA, proteomic, and metabolomic methodology (Roher et al., 2009; Maarouf et al., 2013; Lewitt et al., 2013; Maarouf et al., 2012; Burgos et al., 2014; Janelidze et al., 2015). A comprehensive proteomics study comparing ventricular and lumbar CSF from the same individuals observed significant differences in protein levels for only two proteins (Simonson et al., 2010).

The points above suggest that *postmortem* CSF SAA results may be broadly comparable to *antemortem* results, though more research is needed to fully understand differences across disease stages. We have therefore added a discussion on the differences:

Discussion section, pages 10-11:

“... Finally, while the use of postmortem CSF samples offers the advantage of collection at the same point in time as the neuropathological assessment, several potential limitations should be considered. First, it has been speculated that postmortem CSF may be affected by overall instability or protein degradation. However, it should be noted that the current cohort has drawn CSF samples from subjects with a short postmortem interval (PMI; median=3.1 hours). Multiple studies, including our own, have demonstrated that postmortem CSF yields results comparable to antemortem CSF regarding analyses such as Western blot, ELISA, proteomic, and metabolomic methodology³³⁻³⁸. Second, concerns have been raised about the equivalence of ventricular versus lumbar CSF. However, a comprehensive proteomics study comparing ventricular and lumbar CSF from the same individuals observed significant differences in protein levels for only two proteins³⁹. Third, prior studies using autopsy-confirmed antemortem CSF α -syn SAA testing reported sensitivity and specificity values for detecting LBP that are similar to those observed in the current study^{3,19-28}. Notably, one study directly compared the

sensitivity and specificity of CSF α -syn SAA testing using both antemortem and postmortem samples from the same individuals, and found overall sensitivity to be slightly higher in postmortem CSF (80% vs. 71.2%), possibly because postmortem CSF was collected closer in time to autopsy than antemortem CSF, while specificity was slightly higher in antemortem CSF (98.1% vs. 88.5%)²². While these findings suggest that postmortem CSF SAA results may generally be comparable to those obtained from antemortem samples, further research is needed to fully understand the differences.”

In response to the reviewer’s suggestion, we have also included new analyses assessing the performance of the two-step workflow in individuals from the PPMI cohort with *in vivo* CSF samples. Briefly, we selected all individuals with available UPSIT scores and CSF α -syn SAA testing, excluding those with a genetic PD mutation and those who were recruited into the prodromal cohort, as their inclusion was based on hyposmia, to avoid bias introduced by UPSIT-driven recruitment. The demographics of the resulting 1209 individuals is shown in Table S10 (see below).

Table S10. PPMI cohort characteristics at baseline

	Whole cohort (N=1209)
Age, years	65.1 (8.7)
Sex, n female	566 (46.8%)
UPSIT score	23.9 (8.2)
CSF SAA+, n	832 (68.8%)
Primary clinical diagnosis	
Idiopathic PD	674 (55.7%)
Alzheimer’s disease	2 (0.2%)
Corticobasal syndrome	1 (0.1%)
Dementia with Lewy bodies	5 (0.4%)
Essential tremor	20 (1.7%)
Juvenile autosomal recessive parkinsonsim	1 (0.1%)
Control	470 (38.9%)
Spinocerebellar Ataxia	1 (0.1%)
Other neurological disorder(s)	1 (0.1%)

CSF = cerebrospinal fluid; LBP = Lewy body pathology; PD = Parkinson’s disease; SAA = Seed Amplification Assay; UPSIT = University of Pennsylvania Smell Identification Test.

As the gold standard of *postmortem* neuropathology was not available in the PPMI cohort, we applied the UPSIT-, age-, and sex-based logistic regression model that was trained on the full autopsy cohort, to the PPMI cohort, to categorize individuals into low and high risk groups for cortical LBP. We then estimated the performance of the UPSIT-based risk classification for predicting CSF α -syn SAA status, and the reduction of required CSF tests by using the UPSIT

as a prescreening measure (see Figure 3 below). The UPSIT-based risk classification could accurately predict CSF SAA status, with an accuracy of 79% (95% CI = 77%-81%), a PPV of 82% (95% CI = 80%-85%), and a NPV of 69% (64%-74%). These findings demonstrate that the two-step workflow also performs well in a clinical *in vivo cohort*.

Figure 3. Performance of the two-step workflow in an *in vivo* clinical cohort

The two-step workflow applied to an *in vivo* clinical cohort. (a) Distribution of model-derived probabilities for cortical Lewy body pathology (LBP) based on a logistic regression model including UPSIT scores, age, and sex as predictors, trained on the autopsy dataset. A probability threshold corresponding to 95% sensitivity, derived from the autopsy dataset, was used to classify individuals as low (blue dots) or high (red dots) risk for cortical LBP. (b) Performance metrics of the two-step model for predicting CSF α -syn SAA status, including accuracy, positive predictive value (PPV), and negative predictive value (NPV), with 95% bootstrap confidence intervals. (c) Reduction in the number of CSF tests required using the two-step workflow, with 95% bootstrap confidence intervals.

We have added these additional analyses to the manuscript:

Abstract, page 3:

“... The workflow demonstrated strong generalizability in an independent *in vivo* clinical cohort ($N=1209$; mean age= 65.1 ± 8.7 years; 46.8% female) where it accurately predicted CSF α -syn SAA status (accuracy=79%) and reduced the number of required CSF tests by 26%.”

Introduction, page 5:

“... To confirm that the use of postmortem CSF did not introduce bias, we further validated the two-step workflow in an independent cohort with *in vivo* CSF α -syn SAA data ($N=1209$).”

Methods, page 12:

“... Detailed information about the PPMI study (ClinicalTrials.gov, number NCT01141023) can be found on the PPMI website (<https://www.ppmi-info.org>) and in previous publications^{40,41}. We included all individuals with UPSIT scores and CSF α -syn SAA testing, excluding those with a genetic PD mutation and those who were recruited into the prodromal cohort, as their inclusion was based on hyposmia, to avoid bias introduced by UPSIT-driven recruitment ($N=1209$). CSF α -syn SAA testing was described in detail previously^{42,43}.”

Methods, page 16:

“... Finally, we applied the UPSIT-, age-, and sex-based logistic regression model that was trained on the full AZSAND/BBDP cohort, to the PPMI cohort, to categorize individuals into low- and high-risk groups for cortical LBP. We then estimated the performance (accuracy, PPV, and NPV) of the UPSIT-based risk classification for predicting CSF α -syn SAA status, and the reduction of required CSF tests by using the UPSIT as a prescreening measure.”

Results, page 8:

“... To examine whether the use of postmortem CSF influenced the results, we also validated the two-step workflow in an independent cohort (i.e., the Parkinson’s Progression Markers Initiative [PPMI] cohort) consisting of 1209 individuals with *in vivo* CSF α -syn SAA and UPSIT available. Mean age was 65.1±8.7 years, 46.8% was female, and 68.8% was CSF α -syn SAA+ (Table S10). The majority had a PD diagnosis (55.7%) or was a healthy control (38.9%) (Table S10). We applied the UPSIT-based risk stratification model from the autopsy cohort to the *in vivo* cohort, where it yielded robust predictions of CSF α -syn SAA status (accuracy=79%; PPV=82%; NPV=69%), while reducing the number of CSF tests required by 26% (Figure 3).”

The authors do not give the interval between death and autopsy.

Authors’ response:

The average time interval between death and autopsy was 3.9 hours (SD=3.9 hours), as described in Supplementary Table 1. We have now added this information to the main manuscript, Results section, page 5:

“... The mean age at death was 86.2±7.8 years, 42.7% was female, mean postmortem interval (PMI) was 3.9±3.9 hours, and the mean interval between UPSIT test and death was 3.2±2.3 years (Table S1).”

The UPSIT alone already showed an accuracy of 80% the Seed Aggregation assay increases it to 95%.

There are already substantial studies showing a high accuracy of the Aggregation assay in CSF, that has not been collected by autopsy and that still have additional smell testings done.

Authors’ response:

We thank the reviewer for their comment. We recognize that *in vivo* CSF α -syn SAA testing has been explored previously. However, the novelty of the current study lies in combining *antemortem* olfactory testing with CSF SAA testing and neuropathological confirmation in a large, well-characterized cohort. This unique integration of *in vivo* and *postmortem* data provides the opportunity to validate the proposed two-step diagnostic workflow, i.e., combining a simple clinical screening tool with a biomarker assay, against the gold standard of neuropathology. In addition, the current cohort includes a broad range of clinical diagnoses beyond Lewy body disease, allowing for robust evaluation of this approach across different clinical groups.

Response to reviewers

We thank the Editor and reviewers for taking the time to review our manuscript and providing constructive feedback. We have revised the manuscript accordingly. Please find below the original comments from reviewers in black and our corresponding responses in blue.

Reviewer #1

The authors have done a very good job of responding to my comments and the comments of others.

Authors' response:

We thank the reviewer for their positive feedback and for taking the time to review our work. Below, please find a detailed response to each comment.

I have a few minor things to add here:

1. The authors have added sections to justify the use of post-mortem CSF and showing that their use may be comparable to ante-mortem CSF in terms of test characteristics and offer the benefit of being most proximal to neuropathological assessment. Can the authors state if they centrifuged samples or discarded overly bloody samples or those with visible tissue pieces in the samples? Other groups have done this.

Authors' response:

All samples were centrifuged after sample collection. Specifically, CSF was ejected into 15 mL disposable polypropylene tubes and centrifuged at 2000 g for 10 minutes. Supernatants were aliquoted into 0.5 mL polypropylene microcentrifuge tubes and stored at 80 degrees Celsius. Bloody samples were discarded ($n=12$). We have added this information to the Methods section, page 12:

"...We included all autopsied BBDP participants with antemortem UPSIT scores and postmortem ventricular brain CSF of sufficient quality ($N=358$), excluding 12 samples with visible blood contamination."

2. If logistic regression models were used that included UPSIT scores, age, and sex to predict likelihood of SAA positivity, it would be good to document the full models in the supplement to be able to see how much age and sex contributed to the findings. To a certain extent, it may help with the limitation that UPSIT scores were used instead of the age/sex adjusted percentiles. The authors points about derivations of normative values is taken but using the percentiles would make this approach easier to apply in the clinic.

Authors' response:

We have now included the full logistic regression model including UPSIT scores, age, and sex in the Supplementary Materials, as shown below.

In addition, we have added the following sentence to the Discussion section, page 11, to address the reviewer's comment:

"... In addition, future research should explore the use of age- and sex-adjusted UPSIT percentiles to improve clinical applicability."

Table S4. Results of the logistic regression model predicting postmortem cortical LBP

	St. Estimate	Std. error	Z value	P value
Intercept	10.40	1.76	5.91	<.001
UPSIT score	-0.21	0.02	-8.73	<.001
Age at time of UPSIT	-0.09	0.02	-4.29	<.001
Sex (female)	-0.39	0.32	-1.20	0.23

Data correspond to the model across 1000 iterations that yielded median accuracy. Estimates are standardized.

3. The inclusion of the PPMI cohort is impressive and a large undertaking which I appreciate. The authors though should expressly state that PPMI uses a different SAA assay than the one used in the primary analysis and cite appropriate literature if it exists to support the equivalence of the two assays (Kang et al 2019 was a comparative study like this but the assays were different [or different versions]).

Authors' response:

We thank the reviewer for this insightful comment and agree that this is an important point that should be explicitly addressed. A paper by Russo et al. (*Acta Neuropathologica Communications*, 2021) directly compared the two assays, demonstrating remarkable similar CSF SAA results across the same subjects. In addition, the two assays have demonstrated similar performance in CSF cohorts from patients with *postmortem* neuropathological assessments (Bentivenga et al., *Acta Neuropath* 2024; Arnold et al., *Ann Neurol*, 2022; Samudra et al., *Alz Dem*, 2024; Tosun et al., *Alz Dem*, 2024).

We have added this information to the Methods section, page 13:

"... Importantly, while different CSF SAAs were used in the PPMI and AZSAND/BBDP cohorts, a previous study directly comparing the two assays in the same individuals demonstrated remarkably similar results (Russo et al., 2021), consistent with findings from other studies comparing different assay versions (Kang et al., 2019)."

Reviewer #2

I have no further comments.

Authors' response:

We thank the reviewer for their time and valuable feedback in the previous round.

Reviewer #3

The authors have extensively added further analyses and discussions in the revised version of their manuscript, that supports the outcome of the study. I understand that the big advantage is the postmortal analysis of the brains, but the major concern, that the CSF samples have been collected postmortally even with a short postmortal interval remains. postmortally collected CSF can not be compared to antemortem CSF for which the SAA method has been established.

Authors' response:

We thank the reviewer for the opportunity to further elaborate on the use of *postmortem* CSF samples in our study. We think that the use of *postmortem* CSF in this context is justified for the following reasons:

- 1) Our *postmortem* CSF SAA results align closely with those from studies using *antemortem* CSF and autopsy-confirmed pathology (e.g. Arnold et al., 2022; Bentivenga et al., 2024).
- 2) A direct comparison of SAA performance in *antemortem* and *postmortem* CSF within the same individuals showed largely concordant results, with slightly higher sensitivity for *postmortem* samples (Arnold et al., 2022).
- 3) The short *postmortem* interval in our study (median = 3.1 hours) minimizes potential concerns related to protein degradation, and previous studies support the comparability of *postmortem* and *antemortem* CSF in various biomarker assays (e.g. Roher et al., 2009; Maarouf et al., 2013; Lewitt et al., 2013; Maarouf et al., 2012; Burgos et al., 2014; Janelidze et al., 2015).
- 4) To further address this concern, we added new analysis in an independent *in vivo* clinical cohort (PPMI), where the two-step workflow yielded good performance, supporting the generalizability of our findings beyond *postmortem* samples.

Please note that Reviewer 1 and Reviewer 2 raised similar concerns about the use of *postmortem* CSF samples in the first round of revisions but found our rationale to be satisfactory. To further accommodate this reviewer comment, we now more extensively discuss the use of *postmortem* CSF samples in the revised manuscript and report the new analyses throughout the manuscript.

E.g. Results section, page 12:

“... To examine whether the use of *postmortem* CSF influenced the results, we also validated the two-step workflow in an independent cohort (i.e., the Parkinson’s Progression Markers Initiative [PPMI] cohort) consisting of 1209 individuals with *in vivo* CSF α -syn SAA and UPSIT available. Mean age was 65.1 ± 8.7 years, 46.8% was female, and 68.8% was CSF α -syn SAA+ (**Table S11**). The majority had a PD diagnosis (55.7%) or was a healthy control (38.9%) (**Table S11**). We applied the UPSIT-based risk stratification model from the autopsy cohort to the *in vivo* cohort, where it yielded robust predictions of CSF α -syn SAA status (accuracy=79%; PPV=82%; NPV=69%), while reducing the number of CSF tests required by 26% (**Figure 3**).”

E.g. Discussion section, pages 11-12:

“... This study also has several limitations. First, replication in independent cohorts with *postmortem* validation is needed, although there are very few cohorts in the world featuring *antemortem* smell testing, CSF sampling, and detailed *postmortem* neuropathological assessments. Second, the interval between *antemortem* UPSIT testing and autopsy was relatively long. However, analyses limited to participant with a maximum five-year interval did not affect the performance of the two-step workflow. Finally, while the use of *postmortem* CSF samples offers the advantage of collection at the same point in time as the neuropathological assessment, several potential limitations should be considered. First, it has been speculated that *postmortem* CSF may be affected by overall instability or protein degradation. However, it should be noted that the current cohort has drawn CSF samples from subjects with a short *postmortem* interval (PMI; median=3.1 hours). Multiple studies, including our own, have demonstrated that *postmortem* CSF yields results comparable to *antemortem* CSF regarding analyses such as Western blot, ELISA, proteomic, and

metabolomic methodology³³⁻³⁸. Second, concerns have been raised about the equivalence of ventricular versus lumbar CSF. However, a comprehensive proteomics study comparing ventricular and lumbar CSF from the same individuals observed significant differences in protein levels for only two proteins³⁹. Third, prior studies using autopsy-confirmed antemortem CSF α -syn SAA testing reported sensitivity and specificity values for detecting LBP that are similar to those observed in the current study^{3,19-28}. Notably, one study directly compared the sensitivity and specificity of CSF α -syn SAA testing using both antemortem and postmortem samples from the same individuals, and found overall sensitivity to be slightly higher in postmortem CSF (80% vs. 71.2%), possibly because postmortem CSF was collected closer in time to autopsy than antemortem CSF, while specificity was slightly higher in antemortem CSF (98.1% vs. 88.5%)²². Importantly, we show that the use of postmortem CSF did not substantially affect the results, as the two-step workflow accurately predicted CSF SAA status in an independent in vivo cohort. While these findings suggest that postmortem CSF SAA results may generally be comparable to those obtained from antemortem samples, further research is needed to fully understand the differences.”